# De novo design of modular peptide-binding proteins by superhelical matching

Kejia Wu[1,2,3,12], Hua Bai[1,2,4,12], Ya-Ting Chang[5], Rachel Redler[5], Kerrie E. McNally[6], William Sheffler[1,2], T. J. Brunette[1,2], Derrick R. Hicks[1,2], Tomos E. Morgan[6], Tim J. Stevens[6], Adam Broerman[1,2,7], Inna Goreshnik[1,2], Michelle DeWitt[1,2], Cameron M. Chow[1,2], Yihang Shen[8], Lance Stewart[1,2], Emmanuel Derivery[6✉], Daniel Adriano Silva[1,2,9,10✉], Gira Bhabha[5], Damian C. Ekiert[5,11] & David Baker[1,2,4✉]

General approaches for designing sequence-specific peptide-binding proteins would have wide utility in proteomics and synthetic biology. However, designing peptide-binding proteins is challenging, as most peptides do not have defined structures in isolation, and hydrogen bonds must be made to the buried polar groups in the peptide backbone[1–3]. Here, inspired by natural and re-engineered protein–peptide systems[4–11], we set out to design proteins made out of repeating units that bind peptides with repeating sequences, with a one-to-one correspondence between the repeat units of the protein and those of the peptide. We use geometric hashing to identify protein backbones and peptide-docking arrangements that are compatible with bidentate hydrogen bonds between the side chains of the protein and the peptide backbone[12]. The remainder of the protein sequence is then optimized for folding and peptide binding. We design repeat proteins to bind to six different tripeptide-repeat sequences in polyproline II conformations. The proteins are hyperstable and bind to four to six tandem repeats of their tripeptide targets with nanomolar to picomolar affinities in vitro and in living cells. Crystal structures reveal repeating interactions between protein and peptide interactions as designed, including ladders of hydrogen bonds from protein side chains to peptide backbones. By redesigning the binding interfaces of individual repeat units, specificity can be achieved for non-repeating peptide sequences and for disordered regions of native proteins.

A number of naturally occurring protein families bind to peptides with repeating internal sequences[7,9]. The armadillo-repeat proteins, which include the nuclear import receptors, bind to extended peptides with lysine- and arginine-rich sequences such that each repeat unit in the peptide fits into a repeat unit or module in the protein[5,8]. Previous studies have shown that the specificity of individual protein repeat units can be re-engineered, which enables broader recognition of peptide sequences[6,11,13,14]. Although this approach is powerful, it is limited to binding peptides in backbone conformations that are compatible with the geometry of the armadillo repeat. Tetratricopeptide-repeat proteins bind to peptides with a variety of sequences and conformations with lower (micromolar) affinity (for exceptions, see refs. 15–17) and with deviations in each peptide–protein interaction register, which complicates engineering for more general peptide recognition[4,9,10].

## Design approach

We set out to generalize peptide recognition by modular repeat-protein scaffolds to arbitrary repeating-peptide backbone geometries. This requires solving two main challenges: first, building protein structures with a repeat spacing and orientation matching that of the target peptide conformation; and, second, ensuring the replacement of peptide–water hydrogen bonds in the unbound state with peptide–protein hydrogen bonds in the bound state. The first challenge is crucial for modular and extensible sequence recognition: if individual repeat units in the protein are to bind individual repeat units on the peptide in the same orientation, the geometric phasing of the repeat units on protein and peptide must be compatible. The second challenge is important for achieving a high binding affinity: in conformations other than the α- and $3_{10}$-helix, the NH and C=O groups make hydrogen bonds with

[1]Department of Biochemistry, University of Washington, Seattle, WA, USA. [2]Institute for Protein Design, University of Washington, Seattle, WA, USA. [3]Biological Physics, Structure and Design Graduate Program, University of Washington, Seattle, WA, USA. [4]Howard Hughes Medical Institute, University of Washington, Seattle, WA, USA. [5]Department of Cell Biology, New York University School of Medicine, New York, NY, USA. [6]MRC Laboratory of Molecular Biology, Cambridge, UK. [7]Department of Chemical Engineering, University of Washington, Seattle, WA, USA. [8]Department of Computational Biology, Carnegie Mellon University, Pittsburgh, PA, USA. [9]Division of Life Science, The Hong Kong University of Science and Technology, Kowloon, Hong Kong. [10]Monod Bio, Seattle, WA, USA. [11]Department of Microbiology, New York University School of Medicine, New York, NY, USA. [12]These authors contributed equally: Kejia Wu, Hua Bai. ✉e-mail: derivery@mrc-lmb.cam.ac.uk; dadriano@gmail.com; dabaker@uw.edu

water in the unbound state that need to be replaced with hydrogen bonds to the protein upon binding to avoid incurring a substantial free-energy penalty[15].

To address the first challenge, we reasoned that a necessary (but not sufficient) criterion for in-phase geometric matching between repeating units on the designed protein and repeating units on the peptide was a correspondence between the superhelices that the two trace out. All repeating polymeric structures trace out superhelices that can be described by three parameters: the translation (rise) along the helical axis per repeat unit; the rotation (twist) around this axis; and the distance (radius) of the repeat unit centroid from the axis[18,19] (Fig. 1a). We generated large sets of repeating-protein backbones that sampled a wide range of superhelical geometries (see Methods). We then generated corresponding sets of repeating-peptide backbones by randomly sampling di-peptide and tri-peptide conformations (avoiding intra-peptide steric clashes), and then repeating these 4–6 times to generate 8–18-residue peptides. We then searched for matching pairs of repeat-protein and repeat-peptide backbones, requiring the rise to be within 0.2 Å, the twist to be within 5° and the radius to differ by at least 4 Å (the difference in radius is necessary to avoid clashing between peptide and protein; the peptide can wrap either outside or inside the protein).

To address the second challenge, we reasoned that bidentate hydrogen bonds between side chains on the protein and pairs of backbone groups or backbone and side-chain groups on the peptide could allow the burying of sufficient peptide surface area on the protein to achieve high-affinity binding without incurring a large desolvation penalty[20,21]. As the geometric requirements for such bidentate hydrogen bonds are quite strict, we developed a geometric hashing approach to enable rapid identification of rigid-body docks of the peptide on the protein that are compatible with ladders of bidentate interactions. To generate the hash tables for bidentate side-chain–backbone interactions, Monte Carlo simulations of individual side-chain functional groups making bidentate hydrogen-bonding interactions with peptide backbone and/or side-chain groups were performed using the Rosetta energy function[12], and a move set consisting of both rigid-body perturbations and changes to the peptide backbone torsions (Fig. 1b; see Methods for details). For each accepted (low-energy) arrangement, side-chain rotamer conformations were built backwards from the functional group to identify placements of the protein backbone from which the bidentate interaction could be realized. The results were stored in hash tables: for each placement, a hash key was computed from the rigid-body transformation and the peptide backbone and side-chain torsion angles determining the position of the hydrogen-bonding groups (for example, the phi and psi torsion angles for a bidentate hydrogen bond to the NH and CO groups of the same amino acid), and the chi angles of the corresponding rotamer were stored in the hash for this key[20]. Hash tables were generated for Asn and Gln making bidentate interactions with the N–H and C=O groups on the backbone of a single residue or adjacent residues, for Asp or Glu making bidentate interactions with the N–H groups of two successive amino acids, and for side-chain–side-chain pi–pi and cation–pi interactions (see Methods).

To identify rigid-body docks that enable multiple bidentate hydrogen bonds between the repeat protein and the repeat peptide, we took advantage of the fact that for matching two superhelical structures along their common axis, there are only two degrees of freedom: the relative translation and rotation along this axis. For each repeat protein–repeat peptide pair, we performed a grid search in these two degrees of freedom, sampling relative translations and rotations in increments of around 1 Å and 10° (Fig. 1c). For each generated dock, we computed the rigid-body orientation for each peptide–protein residue pair, and queried the hash tables to rapidly determine whether bidentate interactions could be made; docks for which the number of matches was less than a set threshold were discarded. For the remaining docks, after building the interacting side chains using the chi

angle information stored in the hash, and rigid-body minimization to optimize hydrogen-bond geometry, we used Rosetta combinatorial optimization to design the protein and peptide sequences[22], keeping the residues that were identified in the hash matching fixed, and enforcing sequence identity between repeats in both the peptide and the protein (see Methods).

In initial calculations with unrestricted sampling of peptide conformations, designs were generated with a wide range of peptide conformations. Examples of repeat proteins designed to bind to extended β-strand, polypeptide II and helical peptide backbones, as well as to a range of less canonical structures, are shown in Extended Data Fig. 1a–c. Reasoning that proline-containing peptides would incur a lower entropic cost upon binding than non-proline-containing peptides, we decided to start our experimental characterization with designs containing at least one proline residue; in most of these designs, the peptide backbone is in or near the polyproline II portion of the Ramachandran map. Our design strategy requires matching the twist of the repeat unit of the peptide with that of the protein, and hence choosing a repeat length of the peptide that generates close to a full 360° turn requires less of a twist in the repeat protein. For the polyproline helix, there are roughly three residues per turn, and, probably because of this, we obtained more designs that target three-residue than two-residue proline-containing repeat units. We selected for experimental characterization 43 designed complexes with near-ideal bidentate hydrogen bonds between protein and peptide, favourable protein–peptide interaction energies[12], interface shape complementary[23] and few interface unsatisfied hydrogen bonds[24], and which consistently retained more than 80% of the interchain hydrogen bonds in 20-ns molecular dynamics trajectories.

## Experimental characterization

We obtained synthetic genes encoding the designed proteins with 6×His tags for purification and terminal biotinylation tags for fluorescent labelling, expressed the proteins in *Escherichia coli* and purified them by Ni-NTA chromatography. Out of 49, 30 were monomeric and soluble. To assess binding, the target peptides were displayed on the yeast cell surface[25], and binding to the repeat proteins was monitored by flow cytometry. To obtain readout of the peptide-binding specificity of individual designs, we in parallel used large-scale array-based oligonucleotide synthesis to generate yeast display libraries encoding all two- and three-residue repeat peptides with eight repeat units each, and used fluorescence-activated cell sorting (FACS) followed by Sanger sequencing to identify the peptides recognized by each designed protein. Many of the designs bound peptides with sequences similar to those targeted, but the affinity and specificity were both relatively low, with most of the successes for three-residue repeat units (Extended Data Table 1a).

On the basis of these results, we sought to increase the peptide sequence specificity of the computational design protocol, focusing on the design of binders for peptides with three-residue repeat units. First, we required that each non-proline residue in the peptide make specific contacts with the protein, and that the pockets and grooves engaging side chains emanating from the two sides of the peptide were quite distinct. Second, after the design stage, we evaluated the change in binding energy (Rosetta ddG)[26] for all single-residue changes to the peptide repeating unit, and selected only designs for which the design target sequence made the most favourable interactions with the designed protein. Third, we used computational alanine scanning to remove hydrophobic residues on the protein surface that did not contribute to binding specificity, to decrease non-specific binding[27]. Fourth, to assess the structural specificity of the designed peptide-binding interface, we performed Monte Carlo flexible backbone docking calculations, starting from large numbers of peptide conformations with superhelical parameters in the range of those of the proteins, and selected

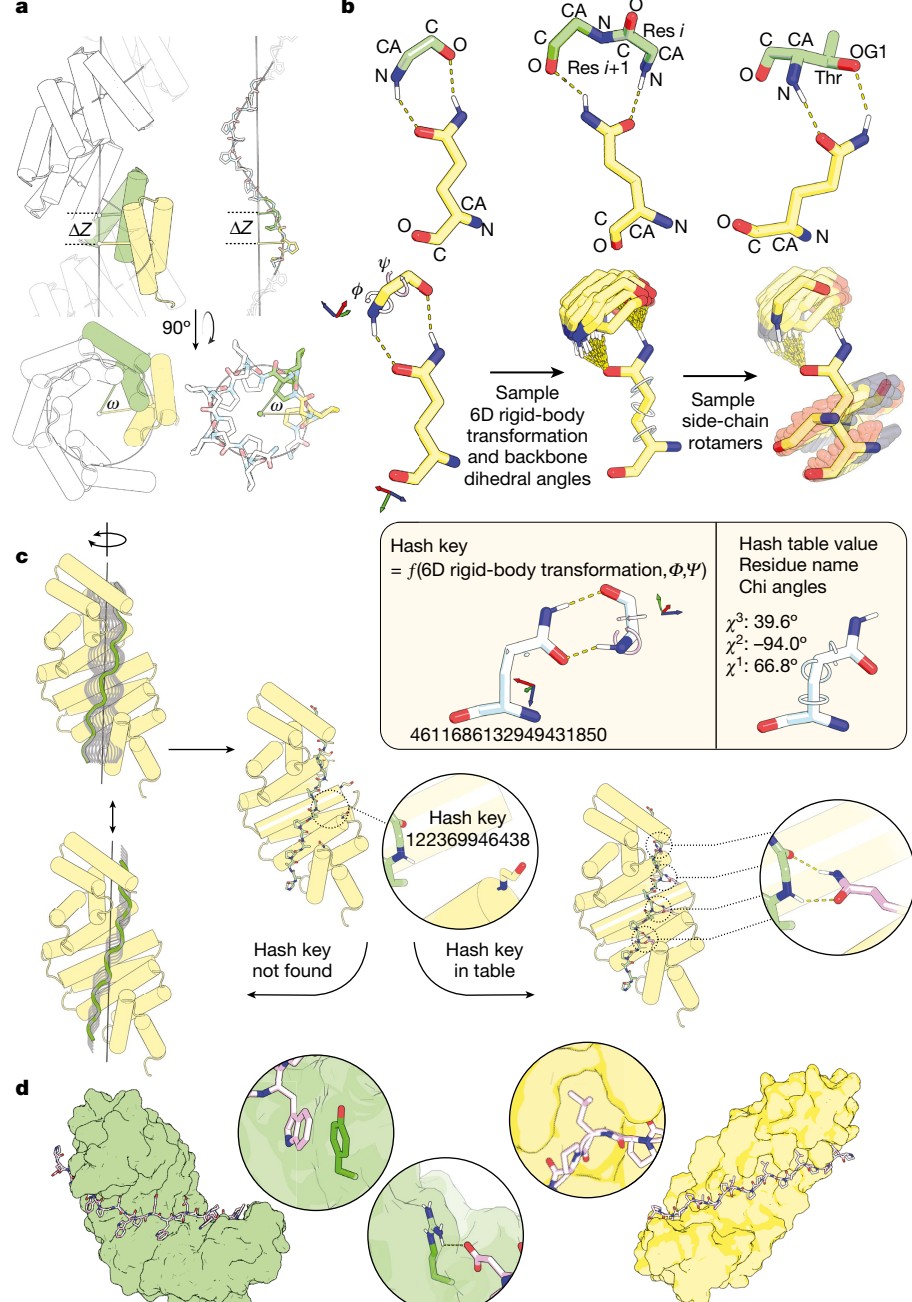

**Fig. 1 | Overview of the procedure for designing modular peptide binders.**
**a**, Like all repeating structures, repeat proteins and peptides form superhelices with constant axial displacement ($\Delta Z$) and angular twist ($\omega$) between adjacent repeat units (shown in green and yellow). For in-register binding, the protein and peptide parameters must match (for some integral multiple of repeat units). **b**, Construction of hash tables for privileged residue–residue interactions. Top row: classes of side-chain–backbone interactions for which hash tables were built. The side-chain amide group of asparagine or glutamine forms bidentate interactions with the N–H and C=O groups on the backbone of a single residue (left) or consecutive residues (middle), or with the backbone N–H group and side-chain oxygen atom of a serine or threonine residue (right). Second row: as illustrated for the case of the glutamine–backbone bidentate interaction, to build the hash table we perform Monte Carlo sampling over the rigid-body orientation between the terminal amide group and the backbone, and the backbone torsions $\varphi$ and $\psi$, saving configurations with low-energy bidentate hydrogen bonds. For each configuration, the possible placements for the backbone of the glutamine are enumerated by growing side-chain rotamers back from the terminal amide. Third row: from the six rigid-body degrees of

freedom relating the backbones of the two residues, together with the two $\varphi$ and $\psi$ torsion angle degrees of freedom, a hash key is calculated using an eight-dimensional hashing scheme. The hash key is then added to the hash table with the side-chain name and torsions as the value. CA, $\alpha$-carbon; OG, $\gamma$-oxygen. **c**, To dock repeat proteins and repeat peptides with compatible superhelical parameters, their superhelical axes are first aligned, and the repeat peptide is then rotated around and slid along this axis. For each of these docks, for each pair of repeat protein–repeat peptide residues within a threshold distance, the hash key is calculated from the rigid-body transform between backbones and the backbone torsions of the peptide residue, and the hash table is interrogated. If the key is found in the hash table, side chains with the stored identities and torsion angles are installed in the docking interface. **d**, The sequence of the remainder of the interface is optimized using Rosetta for high-affinity binding. Two representative designed binding complexes are shown to highlight the peptide-binding groove and the shape complementarity. The magnified views illustrate hydrophobic interactions (right), salt bridges (middle) and π–π stacks (left) incorporated during design.

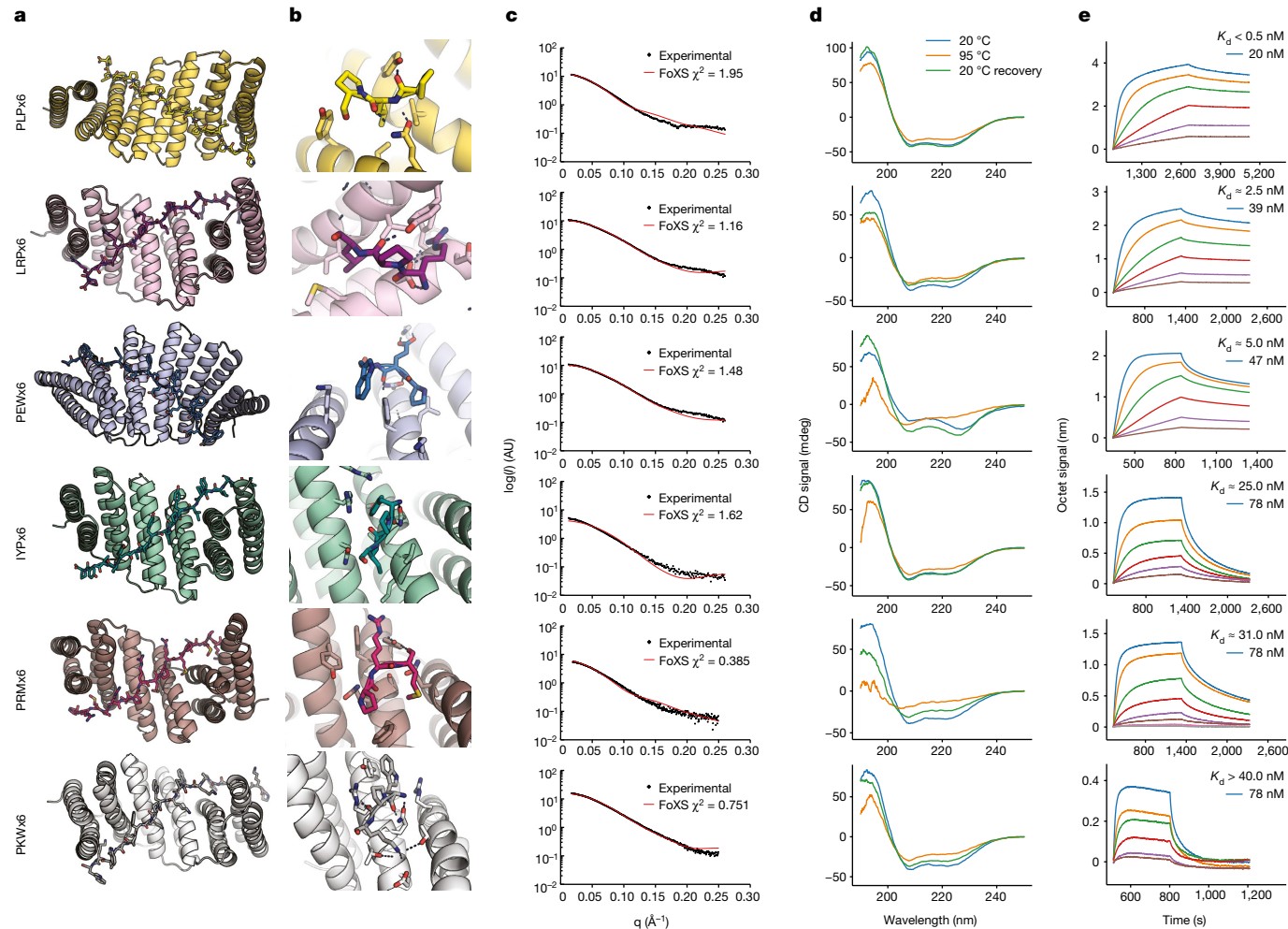

**Fig. 2 | Biophysical characterization of designed protein–peptide complexes. a,** Computational models of the designed six-repeat version of protein–peptide complexes. Designed proteins are shown in cartoons and peptides in sticks. **b,** Magnified views for single designed protein–peptide interaction units. Residues interacting across the interface are shown in sticks. **c,** Predicted SAXS profiles overlaid on experimental SAXS data points. The scattering vector q is on the *x* axis (from 0 to 0.25) and the intensity (*I*) is on the *y* axis on a logarithmic scale. AU, arbitrary units. **d,** Circular dichroism (CD)

spectra at different temperatures (blue, 20 °C; orange, 95 °C; green, 95 °C followed by 20 °C). **e,** Bio-layer interferometry characterization of the binding of designed proteins to the corresponding peptide targets. Twofold serial dilutions were tested for each binder and the highest concentration is labelled. The biotinylated target peptides were loaded onto streptavidin biosensors, and incubated with designed binders in solution to measure association and dissociation.

those designs with converged peptide backbones (root-mean-square deviation (RMSD) < 2.0 between the 20 lowest ddG designs) close to the design model (RMSD < 1.5) (Extended Data Fig. 1d).

We tested 54 second-round designed protein–peptide pairs using the yeast flow cytometry assay described above. Forty-two of the designed proteins were solubly expressed in *E. coli*, and 16 bound their targets with considerably higher affinity and specificity than in the first round (Extended Data Table 1b). We selected six designs with diverse superhelical parameters and shapes, and a range of target peptides for more detailed characterization (Fig. 2). As evident in the design models (Fig. 2a), there is a one-to-one match between the six repeat units in the protein and in the target peptide (Fig. 2b shows a single unit interaction). Small-angle X-ray scattering (SAXS) profiles[28,29] were close to those computed from the design models, suggesting that the proteins fold into the designed shapes in solution (Fig. 2c and Extended Data Table 2b). Circular dichroism studies showed that all six were largely helical and thermostable up to 95 °C (Fig. 2d). Bio-layer interferometry characterization of binding to biotinylated target peptides immobilized on Octet sensor chips revealed $K_d$ values ranging from less than 500 pM (below the instrument level of detection) to around 40 nM; five out

of six had a dissociation half-life of at least 500 s, and for three of the six there was little dissociation after 2,000 s (Fig. 2e; little decrease in binding was observed after storage of the proteins for 30 days at 4 °C, Extended Data Fig. 2). The binding surfaces of several related designs were subjected to site-saturation mutagenesis (SSM)[30] on yeast, and after the incorporation of one to three enriched substitutions, binding was observed by flow cytometry using only 10 pM biotinylated cognate peptide (Extended Data Fig. 3).

Many cell biology approaches[31] involve tagging cellular target proteins with a protein or peptide, and then introducing into the same cell a protein that binds the tag with high affinity and specificity, but does not bind endogenous targets. A bottleneck in such studies is that binders obtained from antibody scaffold (scFV or VHH)-based library screens often do not fold properly in the reducing environment of the cytosol, resulting in a loss of binding[32]. We reasoned that our binders would not have this limitation as they are designed for stability and lack disulfide bonds. As a proof of concept, we co-expressed the peptide PLPx6 fused to GFP and its cognate binder, RPB_PLP2_R6, a variant of RPB_PLP1_R6, fused to both mScarlet and a targeting sequence for the mitochondrial outer membrane (Fig. 3a). (In the naming convention

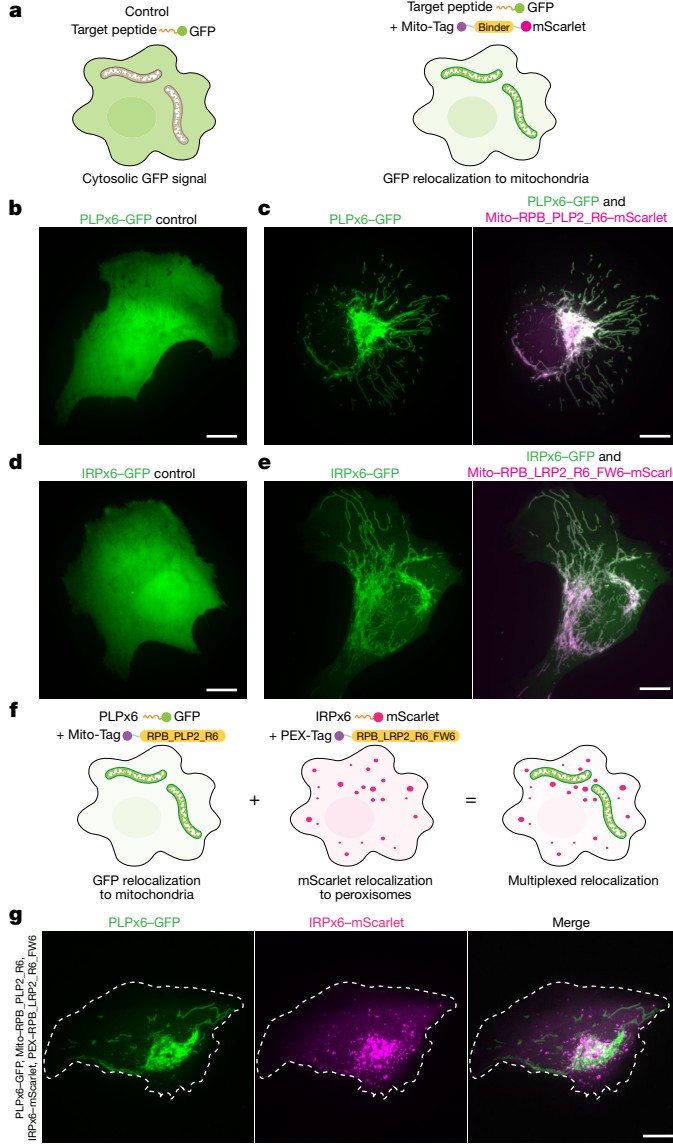

**Fig. 3 | Designed binders function in living cells. a**, Experimental design. U2OS cells co-express the target peptide fused to GFP and a fusion between the specific binder fused to mScarlet and a mitochondria-targeting sequence (Mito-Tag). If binding occurs in cells, the GFP signal is relocalized to the mitochondria, whereas control cells that do not express the binder show a cytosolic GFP signal. **b**–**e**, In vivo binding. Live, spreading U2OS cells expressing PLPx6–GFP alone (**b**), IRPx6–GFP alone (**d**), PLPx6–GFP and Mito–RPB_PLP2_R6–mScarlet (**c**) or IRPx6–GFP and Mito–RPB_LRP2_R6_FW6–mScarlet (**e**) were imaged by spinning disk confocal microscopy (SDCM). Note that the GFP signal is cytosolic in the control but relocalized to the mitochondria after co-expression with the respective binder. **f**,**g**, In vivo multiplexing. **f**, Experimental design. U2OS cells co-express two target peptides, one fused to GFP and the other to mScarlet, and their corresponding specific binder fused to mitochondria- or peroxisome-targeting sequences. If orthogonal binding occurs, GFP and mScarlet signals should not overlap. **g**, Live, spreading U2OS cells co-expressing PLPx6–GFP, IRPx6–mScarlet, Mito–RPB_PLP2_R6 and PEX–RPB_LRP2_R6_FW6 imaged by SDCM. Note the absence of overlap between channels. Images correspond to maximum intensity z-projections (Δz = 6 μm). Dashed line indicates the cell outline. Scale bars, 10 μm.

here and throughout the remainder of the text, 'RPB' indicates 'repeat peptide binder'; 'PLP' indicates the intended peptide specificity (for proline-leucine-proline in this case); '2' indicates the specific module designed to bind this peptide unit; and 'R6' indicates the number

(six) of repeat units. In peptide names, the sequence 'PLP' is followed by the number of repeats 'x6'. In protein–peptide complex descriptors, the protein name is specified first, followed by a dash and then the peptide name.) Although the PLPx6 peptide on its own was diffuse in the cytosol (Fig. 3b), after co-expression with the binder, it was relocalized to the mitochondria (Fig. 3c and Extended Data Fig. 2b). Thus, the PLPx6–RPB_PLP2_R6 pair retains binding activity in cells. Similar results were obtained for IRPx6–GFP and RPB_LRP2_R6_FW6 (Fig. 3d,e).

If individual repeat units on the designed protein engage individual repeat units on the target peptide, the binding affinity should increase when the number of repeats is increased. We investigated this with four of our designed systems—in two cases varying the number of protein repeats while keeping the peptide constant, and in the other two cases, varying the number of peptide repeats while keeping the protein constant. Six-repeat versions of RPB_LRP2_R6 and RPB_PEW2_R6 had a higher affinity for eight-repeat LRP and PEW peptides than did four-repeat versions, without any decrease in specificity (Extended Data Fig. 4a). Similarly, six-repeat IYP and PLP peptides had a higher affinity for six-repeat versions of the cognate designed repeat proteins (RPB_IYP1_R6 and RPB_PLP1_R6) than did four-repeat versions (Extended Data Fig. 4b). These results are consistent with a one-to-one modular interaction between repeat units on the protein and repeat units on the peptide, and suggest that a very high binding affinity could be achieved simply by increasing the number of interacting repeat units. This ability to vary the affinity by varying the number of repeats could be useful in many contexts in which competitive binding would be advantageous. For example, when isolating proteins by affinity purification, a peptide with a larger number of repeats than that fused to the protein being expressed could be used for elution.

## High-resolution structural validation

To assess the structural accuracy of our design method, we used X-ray crystallography. We obtained high-resolution co-crystal structures of three first-round designs (RPB_PEW3_R4–PAWx4, RPB_LRP2_R4–LRPx4, RPB_PLP3_R6–PLPx6) and one second-round design (RPB_PLP1_R6–PLPx6) (Fig. 4); and a crystal structure of the unbound first-round design RPB_LRP2_R4 (Extended Data Fig. 5a; interface side-chain RMSD values for all crystal structures are in Extended Data Table 2a). In the crystal structure of RPB_PLP3_R6–PLPx6, the PLP units fit exactly into the designed curved groove formed by repeating tyrosine, alanine and tryptophan residues, matching the design model with near atomic accuracy (Cα RMSD for protein, protein–peptide interface and full complex: 1.70 Å, 2.00 Å and 1.64 Å, respectively; Fig. 4b and Extended Data Fig. 5b). In the co-crystal structure of RPB_PEW3_R4–PAWx4, as in the design model, the PAW units bind to a relatively flat groove formed by repeating histidine residues and glutamine residues, as designed (Fig. 4a and Extended Data Fig. 5c, RMSD 2.08 Å between design and crystal structure over the protein, median RMSD 2.12 Å over the peptide and interface between crystal and docked peptide ensemble; Extended Data Table 2a). For RPB_LRP2_R4–LRPx4, flexible backbone docking converged with the LRP units fitting in between repeating glutamine residues and phenylalanine residues as designed, and the peptide arginine side chain sampling two distinct states associated with parallel and antiparallel protein-binding modes (Extended Data Fig. 4c). The lowest-energy docked structure was close to the crystal structure, with Cα RMSD values of 1.15 Å, 0.98 Å and 1.16 Å for the protein alone, the peptide plus interface and the entire complex, respectively (Fig. 3c and Extended Data Table 2a). SSM interface footprinting results were consistent with the design model and crystal structure (Extended Data Fig. 6), and an Phe-to-Trp substitution that increases interactions across the interface substantially increased the affinity (Extended Data Fig. 3d).

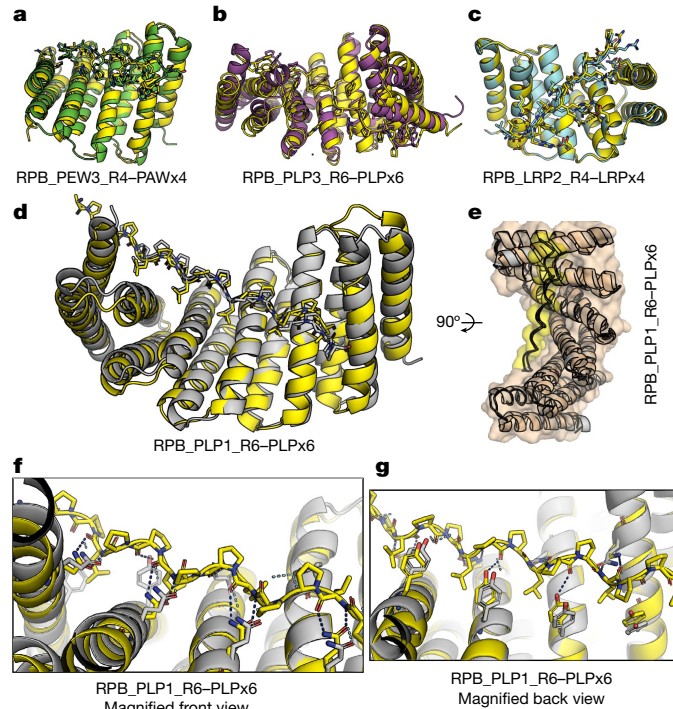

**a**, RPB_PEW3_R4–PAWx4. **b**, RPB_PLP3_R6–PLPx6. **c**, RPB_LRP2_R4–LRPx4.

RPB_PLP1_R6–PLPx6

90°

RPB_PLP1_R6–PLPx6

RPB_PLP1_R6–PLPx6
Magnified front view

RPB_PLP1_R6–PLPx6
Magnified back view

**Fig. 4 | Evaluation of design accuracy by X-ray crystallography.**
**a**–**c**, Superposition of computational design models (coloured) on experimentally determined crystal structures (yellow). **a**, RPB_PEW3_R4–PAWx4. **b**, RPB_PLP3_R6–PLPx6. **c**, RPB_LRP2_R4–LRPx4. **d**–**g**, RPB_PLP1_R6–PLPx6, **d**, Overview of the superimposition of the computational design model and the crystal structure. **e**, A 90° rotation of **d**. The complex is shown in surface mode (protein in orange and peptide in yellow) to highlight the shape complementarity. **f**, Zoom in on the internal three units from **d** (front view). Glutamine residues from the protein in both the design and the crystal structure are shown as sticks to highlight the accuracy of the designed side-chain-to-backbone bidentate ladder. **g**, View from the side opposite to **f**. Tyrosine residues from the protein in both the design and the crystal structure are shown as sticks to highlight the accuracy of the designed polar interactions.

The 2.15-Å crystal structure of the second-round design RPB_PLP1_R6–PLPx6 highlights key features of the computational design protocol. The PLPx6 peptide binds to the slightly curved groove mainly through polar interactions from tyrosine, hydrophobic interactions from valine and side-chain–backbone bidentate hydrogen bonds from glutamine, exactly as designed (Fig. 4d–g; RMSD 1.11 Å for the protein–peptide interface and 1.91 Å for the complex). All interacting side chains from both the protein side and the peptide side in the computational design model are nearly perfectly recapitulated in the crystal structure. This design has near-picomolar binding affinity (Fig. 2d) and high specificity for the PLP target sequence (Fig. 5a).

We next investigated the specificity of the six designs (Fig. 5a). The PLPx6, LRPx6, PEWx6, IYPx6 and PKWx6 binders showed almost complete orthogonality in the concentration range from around 5 nM to 40 nM, with each design binding its cognate designed repeat peptide much more strongly than the other repeat peptides. For example, PLPx6 binds RPB_PLP1_R6 strongly at 5 nM, but shows no binding signal to RPB_IYP1_R6 at 40 nM, whereas PEWx6 binds RPB_PEW1_R6 but not RPB_PKW1_R6 at 20 nM. Some cross-talk was observed between the PRMx6 and LRPx6 binders, perhaps involving the arginine residue, which makes cation–pi interactions in both designs. We observe similar interaction orthogonality in cells: the IRPx6 and PLPx6 binders specifically direct the localization of their cognate peptides to different compartments when co-expressed in the same cells (Fig. 3e,f).

As described thus far, our approach enables the specific binding of peptides with perfectly repeating structures. To go beyond this limitation and enable a much wider range of non-repeating peptides to be targeted, we investigated the redesign of a subset of the peptide-repeat-unit binding pockets to change their specificity. We broke the symmetry in the designed repetitive binding interface by redesigning both protein and peptide in one or more repeats of six-repeat complexes; the rest of the interface was kept untouched to maintain the binding affinity. After redesign, the peptide backbone conformation was optimized by Monte Carlo resampling and rigid-body optimization (see Methods). Designs were selected for experimental characterization as described above, favouring those for which the new design had a lower binding energy for the new peptide than the original peptide.

We redesigned the PLPx6 binder RPB_PLP3_R6 to bind two PEP units in the third and fourth positions (target binding sequence PLPPLPPEP-PEPPLPPLP or, more concisely, $PLP_2PEP_2PLP_2$). The redesigned protein, called RPB_hyb1_R6, bound the redesigned peptide considerably more tightly in Octet experiments, whereas the original design favoured the previous perfectly repeating sequence, resulting in nearly complete orthogonality (Fig. 5b). We next designed another hybrid starting from the RPB_IYP1_R6–IYPx6 complex, in which we changed three of the IYP units to RYP to generate $IYP_3RYP_3$, and redesigned the corresponding binding pockets. The new design, RPB_hyb2_R6, selectively bound the intended cognate target as well (Fig. 5b). We measured the binding of all four proteins against all four peptides, and observed high specificity of the designed repeat proteins for their intended peptide targets (Fig. 5b).

## Generalization to native disordered regions

The ability to design hybrid binders against non-repetitive sequences opens the door to the de novo design of binders against endogenous proteins. Intrinsically disordered regions have been very difficult to specifically target using other approaches, but are in principle good targets, because binding is not complicated by folding. As a proof of concept, we focused on human ZFC3H1, a 226-kDa protein that together with MTR4 forms the heterotetrameric poly(A) tail exosome targeting (PAXT) complex, which directs a subset of long polyadenylated poly(A) RNAs for exosomal degradation[33,34] (Fig. 6a). We designed binders against ZFC3H1 residues 594–620 ($PLP_4PEDPEQPPKPPF$), which lie within an approximately 100-residue disordered region (Fig. 6a), by extending both the protein and the peptide in the PLPx4 designed complex. On the peptide side, we kept the (PLP)x4 backbone fixed, and used Monte Carlo sampling with Ramachandran map biases to model the remaining sequence (PEDPEQPPKPPF); on the protein side, we extended the PLPx4 design with four additional repeats, designed binding interactions with each peptide conformer and selected eight designs for experimental characterization. These eight designs were expressed, and seven were found to bind the extended target peptide by bio-layer interferometry (Extended Data Fig. 7a). The two highest-affinity designs—αZFC-high and αZFC-low—were found by fluorescence polarization to have $K_d$ values of less than 200 nM and around 1.2 μM, respectively (Fig. 6b,c), somewhat weaker than the synthetic constructs described above. Nevertheless, αZFC-high co-eluted with a 103-amino-acid segment of the disordered region of ZFC3H1 containing the targeting sequence by size-exclusion chromatography (SEC) (Fig. 6d), demonstrating that the binder can recognize the target peptide in the context of a larger protein. αZFC-high specifically pulled down the endogenous ZFC3H1 from human cell extracts when assessed by western blot with established antibodies (Fig. 6e, top), whereas αZFC-low—which has a similar size and surface composition—did not; αZFC-low hence provides a control for non-specific association (see Extended Data Fig. 7b for replicates, and Fig. 6f for independent identification of ZFC3H1 by mass spectrometry). Mass spectrometry revealed that MTR4 was enriched in the αZFC-high pull-down, demonstrating

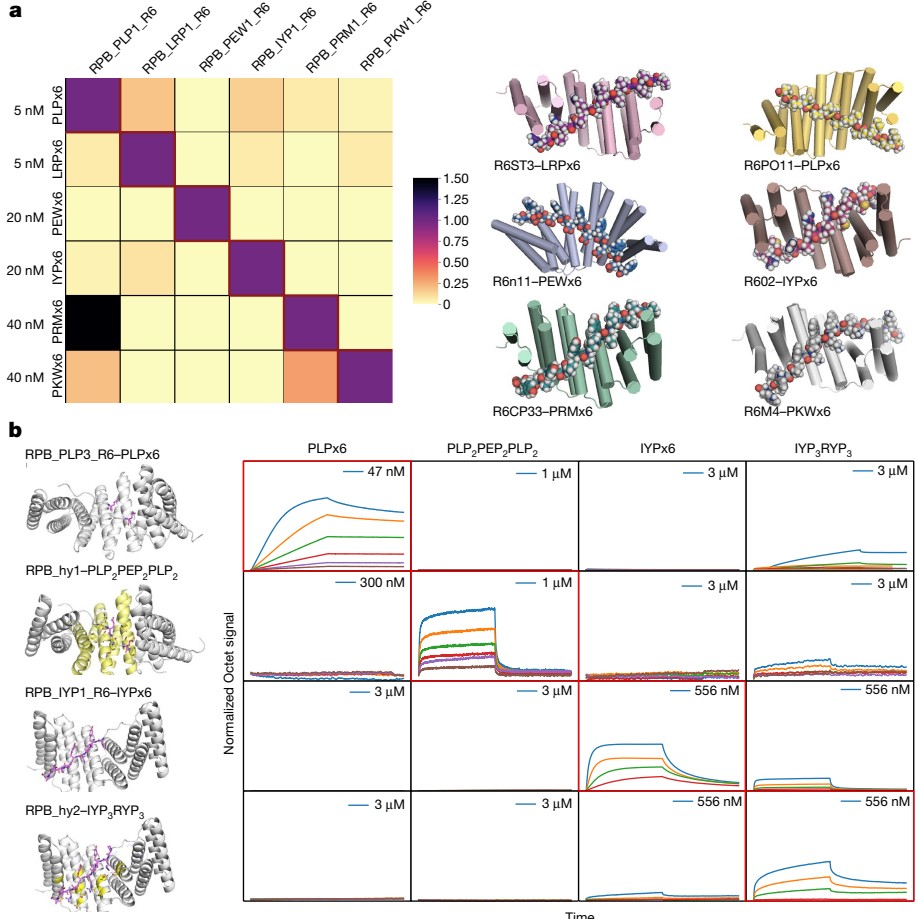

**Fig. 5 | Designed protein–peptide interaction specificity. a**, Left, to assess the cross-reactivity of each designed peptide binder in Fig. 2 with each target peptide, biotinylated target peptides were loaded onto bio-layer interferometry streptavidin sensors and allowed to equilibrate, and the baseline signal was set to zero. The bio-layer interferometry tips were then placed into a solution containing proteins at the indicated concentrations for 500 s and washed with buffer, and dissociation was monitored for another 500 s. The heat map shows the maximum signal for each binder–target pair (cognate and non-cognate) normalized by the maximum signal of the cognate designed binder–target pair. Right, surface shape complementarity of the cognate complexes. The peptides are in sphere representation. **b**, Modular pocket sequence redesign generates binders for peptide sequences that are not strictly repeating. Left, ribbon diagrams of base designs (rows 1 and 3) and versions with a matching subset of the protein and peptide modules redesigned. The ribbon diagrams show the cognate designed and redesigned assemblies; for example, the first row shows a six-repeat PLP binding design in complex with PLPx6, and the second row the same backbone with repeat units 3 and 4 redesigned to bind PEP instead of PLP, in complex with a PLP$_2$PEP$_2$PLP$_2$ peptide. The redesigned peptide and protein residues are shown in purple sticks and yellow, respectively. Right, orthogonality matrix. Biotinylated target peptides were loaded onto biosensors, and incubated with designed binders in solution at the indicated concentrations. Red rectangle boxes indicate cognate complexes. Octet signal was normalized by the maximum signal of the cognate designed binder–target pair.

that the binder can recognize the native PAXT complex in a physiological context. We also detected in the αZFC-high pull-down, but not in the αZFC-low pull-down, other binding partners of ZFC3H1 that are present in the Bioplex 3.0 interactome in multiple cell lines (for example, BUB3 and ZN207)[16–18,35,36], and several RNA-binding proteins that probably associate with PAXT–RNA assemblies (Fig. 6f; see Source Data for the full proteomics dataset).

## Conclusion

Our results show that by matching superhelical parameters between repeating-protein and repeating-peptide conformations, and incorporating specific hydrogen-bonding and hydrophobic interactions between matched protein and peptide repeats, we can now design modular proteins that bind to extended peptides with high affinity and specificity. The strategy should be generalizable to a wide range of repeating-peptide structures, and the ability to break symmetry by redesigning individual repeat units opens the door to more general peptide recognition. Our approach complements existing efforts to achieve general peptide recognition by redesigning naturally occurring repeat proteins; an advantage of our method is that a much broader range of protein conformations and binding-site geometries can be generated by de novo protein design than by starting with a native protein backbone. Proteins that bind to repeating or nearly repeating sequences could have applications as affinity reagents for diseases that are associated with repeat expansions, such as Huntington's disease. Similarly, rigid fusion of protein modules designed to recognize different di-, tri- and tetrapeptide sequences, using the approach described here, provides an avenue to achieving sequence-specific recognition of entirely non-repeating sequences. The ability to design specific binders for proteins that contain large disordered regions—shown here by the specific pull-down of the PAXT complex (Fig. 6)—should help to unravel the functions of this important but relatively poorly understood class of proteins, and should reduce our reliance on animal immunization to generate antibodies, which can also suffer from reproducibility issues. The affinity of around 100 nM that we attained for

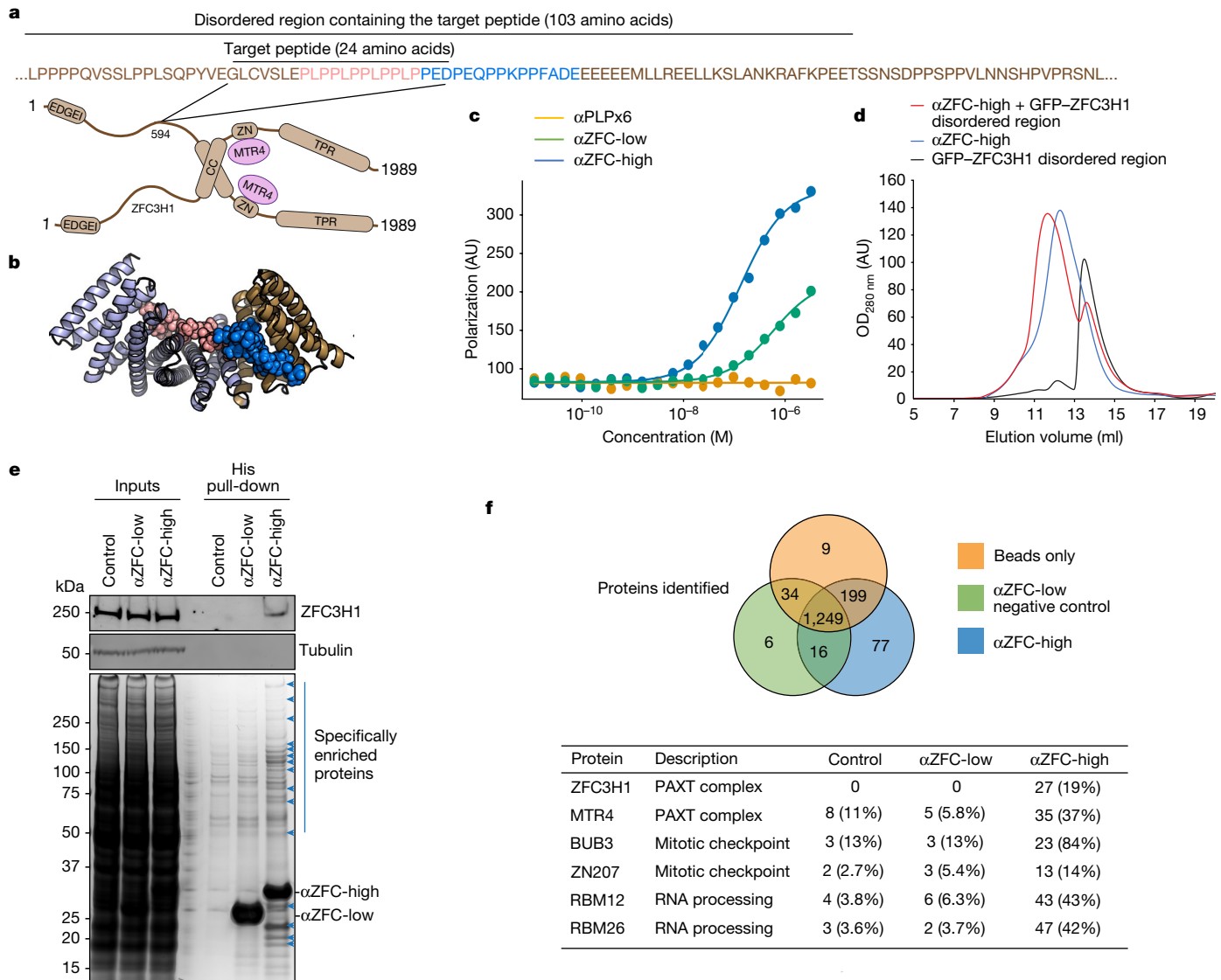

**Fig. 6 | Design of binders to disordered regions of endogenous human proteins. a**, Schematic model of the human PAXT complex composed of a heterotetramer of ZFC3H1 and MTR4. CC, coiled-coil domain; ZN, Zn-finger domain. Inset shows the sequence environment of the target sequence. **b**, Surface shape complementarity between the target peptide from ZFC3H1 (sphere) and the highest-affinity cognate binder, αZFC-high. **c**, Fluorescence polarization binding curves between the indicated ZFC3H1 binders and the target ZFC3H1 peptide (PLP)4PEDPEQPPKPP. As a negative control, we used the (PLP)x6 binder, RPB_PLP3_R6 (see Fig. 4). αZFC-high shows a higher binding affinity to the target peptide than αZFC-low, in contrast with RPB_PLP3_R6, which shows negligible binding. **d**, Superdex 200 10/300 GL SEC profiles of purified αZFC-high, a fusion between GFP and a 103-amino-acid fragment of the disordered region of ZFC3H1 containing the target sequence (see **a**), or a 1:1 mix of the two after two hours of incubation. $OD_{280\,nm}$, optical density at 280 nm. **e**, Top, HeLa cell extracts were subjected to pull-down using the indicated binders bound to Ni-NTA agarose beads, or naked beads as a control. Recovered proteins were processed for western blot against endogenous ZFC3H1 (or tubulin as a loading control). Bottom, Coomassie-stained SDS–PAGE gel of the samples analysed at the top. These panels are representative of $n = 3$ experiments. **f**, Proteomic analysis of the His-pull-down samples shown in **e**. Top, overlap between the proteins identified, setting a threshold of five peptides for correct identification. Bottom, examples of proteins identified (number indicates exclusive peptide count; protein coverage is indicated in parentheses). See Source Data for the full dataset. For gel source data, see Supplementary Fig. 1.

this endogenous binder is compatible with other cellular applications, such as enzyme targeting for specific post-translational modifications in vivo[18,35,36], or for imaging probes, in which a trade-off must always be found between high-affinity interactions for labelling specificity and low-affinity interactions to avoid perturbing protein function[37,38]. More generally, our results reveal the power of computational protein design for targeting peptides and intrinsically disordered regions that do not have rigid three-dimensional structures. Because the designed proteins are expressed at quite high levels and are very stable, we anticipate that these and further designs for a wider range of target sequences will have many uses in proteomics and other applications that require specific peptide recognition.

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

## Methods

### Generation of DHR scaffolds

Each designed helical repeat (DHR) scaffold is formed by a helix-loop-helix-loop topology that is repeated four or more times[18,35,36]. The helices range from 18 to 30 residues and the loops from 3 to 4 residues. The DHR design process goes through backbone design, sequence design and computational validation by energy landscape exploration. To match the peptides, the designs were required to have a twist (omega) between 0.6 and 1.0 radians, a radius of 0 to 13 Å and a rise between 0 and 10 Å. The geometry of a repeat protein can be described by the radius of the super-helix, the axial displacement and the twist[37,38].

The backbone is designed using Rosetta fragment assembly guided by motifs[21]. Backbone coordinates are built up through 3,200 Monte Carlo fragment assembly steps with fragments taken from a non-redundant set of structures from the Protein Data Bank (PDB). After the insertion of each fragment, the rigid-body transform is propagated to the downstream repeats. The score that guides fragment assembly is composed of Van der Waal interactions, packing, backbone dihedral angles and residue-pair-transform (RPX) motifs[21]. RPX motifs are a fast way to measure the full-atom hydrophobic packability of the backbone before assigning side chains. After design, backbones are screened for native-like features. The loops are required to be within 0.4 Å of a naturally occurring loop or rebuilt. Structures with helices above 0.14 Å appear bent and kinked and are discarded. And poorly packed structures in which fewer than four helices are in contact with each other are filtered.

The sequence is designed using Rosetta for each backbone that passes filtering. Design begins in a symmetrical mode in which each repeat is identical using the RepeatProteinRelax mover. Core residues are restricted to be hydrophobic and surface residues hydrophilic using the layer design task operators. Sequence is biased toward natural proteins with a similar local structure using the structure profile mover. After the symmetrical design is complete, the N-terminal and C-terminal repeats are redesigned to eliminate exposed hydrophobics. Designs with poor core packing as measured by Rosetta Holes < 0.5 are then filtered[39].

The designs are computationally validated using the Rosetta ab initio structure prediction on Rosetta@home[40]. Rosetta ab initio verifies that the design is a lower-energy state than the thousands of alternative conformations sampled. Simulating a protein using Rosetta@home can take several days on hundreds of CPUs. To speed this up, we used machine learning to filter designs that were most likely to fail[37,38].

### Backbone generation of curved repeat-protein monomers in polyproline II conformation

A second round of designs was made to ensure that the distance between helices matches the 10.9 Å. distance between prolines in the polyproline II conformation. To design these backbones, we used atom-pair constraints between the first helix of each repeat. The atom-pair constraints were set to 10.9 Å with a tolerance of 0.5 Å. For these designs, we found the topologies that most efficiently produced structures that matched the atom-pair constraints had a helix length of 20 or 21 residues and a loop range of three residues.

### Design of peptide binders

**Modular peptide docking and hashing.** To construct hash tables storing the pre-computed privileged residue interactions, we first surveyed the non-redundant PDB database and extracted the intended interacting residues as seeds. For each seeding interaction residue pair, random perturbations were applied to search for alternative relative conformations of the interacting residues. In the case of the side-chain–backbone bidentate interactions, random rigid-body perturbations were applied to the backbone residues, with a random set of Euler angles drawn from a normal distribution with 0° as the mean

and 60° as the standard deviation, as well as a random set of translation distances in three-dimensional (3D) space drawn from a normal distribution with 0 Å as the mean and 1 Å as the standard deviation. At the same time, the backbone torsion angles $\Phi$ and $\Psi$ of the backbone residue were randomly modified to values drawn from a Ramachandran density plot based on structures from the PDB database. The transformed set of residues losing the intended interactions were discarded. The transformed residues keeping the interactions will be collected. Then, the side chains of the side-chain residues were replaced with all reasonable rotamers, to further diversify the samples of the sets of interacting residues. Finally, the geometry relationship of each set of residues keeping the intended interactions was subjected to an 8D hash function (6D rigid-body transformation plus two torsion angles), and represented with a 64-bit unsigned integer as the key of an entry in the hash table. The identity and the side-chain torsion angles ($X$s) of the side-chain residues were treated as the value of the entry in the hash table. Similar processes were used to build different hash tables for various interactions, with minor alterations. For example, for pi–pi and cation–pi interactions, only a 6D hash function was used, because there is no need for the perturbation and consideration of the backbone torsions. For Asn, Gln, Asp or Glu interacting with two residues on the backbone, a 10D hash table was applied for representing the geometry relationship, and, in these cases, the geometries of the N–H and C=O groups on the backbone were treated as 5D rays.

To sample repeat peptides that match the superhelical parameters of the DHRs, we randomly generate a set of backbone torsion angles $\varphi$ and $\psi$, for example, $[\varphi_1, \psi_1, \varphi_2, \psi_2, \varphi_3, \psi_3]$ for repeats of tripeptide. If any pair of $\varphi$ and $\psi$ angles gets a high Rosetta Ramachandran score above the threshold of −0.5, it means that this pair of torsion angles is likely to introduce intra-peptide steric clashes, and in these cases we randomly regenerate a new pair of $\varphi$ and $\psi$ angles until they are reasonable according to the Rosetta Ramachandran score. Next, we set the backbone torsion angles of the repeat peptide using this set of $\varphi$ and $\psi$ angles repetitively across the eight repeats. And we calculate the superhelical parameters using the 3D coordinates of adjacent repeat units of the repeat peptide. The repeat peptides matching the superhelical parameters of any one of the curated DHRs are saved for the docking step.

To dock cognate repeat proteins and repeat peptides, with matching superhelical parameters, they are first aligned to the *z* axis by their own superhelical axes. In the next step, a 2D grid search (rotation around and translation along the *z* axis) is carried out to sample compatible positions of the repeat peptide in the binding groove of the repeat protein. Once a reasonable dock is generated without steric clash, the relevant hash function is used to iterate through all potential peptide–protein interacting residue sets, to calculate the hash keys. If a hash key exists in the hash table, the interacting side-chain identities and torsion angles will be pulled out immediately and installed on all equivalent positions of this repeat peptide–repeat protein docking conformation. The docked peptide–DHR pair is saved for the interface design step if the peptide–DHR hydrogen-bond interactions are satisfied.

**Design of the peptide-binding interface.** If a single dock was accepted with the designed repetitive peptide–DHR hydrogen bond, the peptide was first trimmed to the exact same repeat number as the DHR (for example, four-repeat or six-repeat). After that, for both peptide and DHR sides, each amino acid was set linked to its corresponding amino acids on the same position in each repeat unit. This was to make sure that all of the following design steps would be carried out with the exact same symmetry inside both the DHR and the peptide.

During our design cycles, the interface neighbour distance is set as 9 Å as the whole designable range around the DHR–peptide binding interface, and 11 Å as the whole minimization range. Three rounds of full hydrophobic FastDesign[21] followed by hydropathic FastDesign were carried out, with each hydrophobic or hydrophilic FastDesign

repeating twice. The Rosetta score function beta_nov16 was chosen in all design cycles. In the produced complex, the peptide itself with an averaged score (three calculations were carried out) larger than 20.0 or a complex score larger than −10.0 were rejected directly.

After the preliminary design was done, we performed two types of sanity check to further optimize the designed peptide sequence, as well as the designed DHR interface. Specifically, for the peptide side, in the tripeptide repeat units, every two amino acids other than proline were scanned for a possible mutation to all twenty amino acids except cysteine, unless a certain originally designed peptide amino acid is making the hashed side-chain–backbone hydrogen bond, or side-chain–side-chain hydrogen bond, or side-chain–side-chain–backbone hydrogen bond with the DHR interface. The DDG (binding energy for the peptide–DHR complex) was compared before and after this peptide side mutation; and the mutation was accepted if the delta DDG (DDG_after − DDG_before) was larger than 1.0. Similarly, we also checked the designed DHR interface by mutation. The whole DHR was scanned. For the designed hydrophobic amino acids that were originally hydrophilic, a delta DDG of −5.0 was set as the threshold to be accepted as a necessary design that made enough binding contribution. For the designed hydropathic amino acids, a delta DDG of −2.0 was used as the threshold.

For experimental characterization, we selected designed complexes with near-ideal bidentate hydrogen bonds between protein and peptide, favourable protein–peptide interaction energies (DDG ≤ −35.0), interface shape complementarity (Iface_SCval ≥ 0.65), tolerable interface unsatisfied hydrogen bonds (Iface_HbondsUnsatBB ≤ 2, Iface_HbondsUnsatSC ≤ 4) and low peptide apo energies (ScoreRes_chainB ≤ 0.9).

**Forward docking.** As for the selected designed complexes from our round-two experiments, forward docking was performed to ensure the specificity in silico. For each designed complex, 10,000 arbitrary peptide conformations were generated as above, using the designed sequence. The same docking protocol was conducted as described in the docking stage, against the untouched designed DHR. FastRelax[41] was then performed for the 10,000 docks, and the DDG versus peptide-backbone RMSD was plotted to check the convergence of the complex. Only the 'converged' complexes were selected for experimental characterization; for example, (i) peptide backbone RMSD < 2.0Å among the top 20 designs with the lowest DDG during forward docking; and (ii) the averaged peptide backbone of the top 20 designs was close to the original design model (RMSD < 1.5 Å).

**Preparation of SSM libraries.** We performed SSM studies for some of the designed peptide–protein binding pairs to gain a better understanding of the peptide-binding modes, and to search for improved peptide binders. For each designed repeat protein, we ordered a SSM library covering the central span of 65 amino acids within the whole repeat protein, owing to the chip DNA size limitation. This span roughly equals one and a half repeating units, across three helices. The chip synthesized DNA oligos for the SSM library were then amplified and transformed to EBY100 yeast together with a linearized pETCON3 vector including the encoding regions of the rest of the designed repeat protein. Each SSM library was subjected to an expression sort first, in which the low-quality sequences due to chip synthesizing defects or recombination errors were filtered out. The collected yeast population, which successfully expresses the designed repeat-protein mutants, will be regrown, and subjected to the next round of peptide-binding sorts. The next-generation sequencing results of this yeast population will also serve as the reference data for SSM analysis. The next round of without-avidity peptide-binding sorts used various concentrations of the target peptide, depending on the initial peptide-binding abilities, ranging from 1 nM to 1,000 nM. The peptide-bound yeast populations were collected and sequenced using the Illumina NextSeq kit. The mutants were identified and compared to the mutants in the expression libraries. Enrichment analysis was used to identify beneficial mutants and provide information for interpreting the peptide-binding modes.

For each mutant, its enrichment value is calculated by dividing its ratio in the peptide-bound population by its ratio in expression population. The enrichment value is then subjected to a $\log_{10}$ transformation, and plotted in heat maps for the SSM analysis.

**Design of binders against endogenous targets.** To evaluate which endogenous proteins could at present be targeted with our method (Fig. 6), we developed Python code to search databases for sub-sequences that match permutations of the set of amino acid triplets for which we designed binders in this study (that is, LRP PEW PLP IYP PKW IRP LRT LRN LRQ RRN PSR PRQ). This code can be accessed freely (https://github.com/tjs23/prot_pep_scan). We then ranked all outputs to find the longest sub-sequence possible, and manually inspected the candidates to find sub-sequences landing in disordered regions. Doing this analysis on the human proteome suggested that ZFC3H1 could be a good target for two main reasons: (1) this protein possesses the sequence (PLP)x4 within a large disordered domain, with downstream sequence (PEDPEQQPPKPPF) within the reach of our binder design method; and (2) this protein is well studied, and—in particular—commercial, highly specific and validated antibodies exist against it.

### Synthetic gene constructs

All genes in this work were ordered from either Integrated DNA Technologies (IDT) or GenScript. For both the first- and the second-round designs, a His tag containing a TEV protease cleavage site and short linkers were added to the N terminus of protein sequences. For the protein lacking a tryptophan residue, a single tryptophan was added to the short N-terminal linker following the TEV protease cleavage site to help with the quantification of protein concentration by A280. The protein sequence along with the linker (MGSSHHHHHH HHSSGGSGGLNDIFEAQKIEWHEGGSGGSENLYFQSG or LEHHHHHH) was reverse-translated into DNA using a custom Python script that attempts to maximize the host-specific codon adaptation index[42] and IDT synthesizability, which includes optimizing whole-gene and local GC content as well as removing repetitive sequences. Finally, a TAATCA stop codon was appended to the end of each gene. Genes were delivered cloned into pET-29b+ between NdeI and XhoI restriction sites. For the second-round designs, the designed amino acid sequences were inserted directly into pET-29b+ between NdeI and XhoI restriction sites.

For the disordered region of ZFC3H1, the 103 amino acids containing the key targeting sequence (LPPPPQVSSLPPLSQPYVEGLCVSLEPLP PLPPLPPLPPEDPEQQPPKPPFADEEEEEEMLLREELLKSLANKRAFKPEETS SNSDPPSPPVLNNSHPVPRSNL) was cloned into a customized vector with sfGFP at the N terminus and His6 at the C terminus with a linker (GGSGSG) in between.

### Protein expression and purification

Proteins were transformed into Lemo21(DE3) *E. coli* from New England Biolabs (NEB) and then expressed as 50-ml cultures in 250-ml flasks using Studiers M2 autoinduction medium with 50 µg ml$^{-1}$ kanamycin. The cultures were either grown at 37 °C for around 6–8 h and then around 18 °C overnight (around 14 h), or at 37 °C for the entire time (around 14 h). Cells were pelleted at 4,000*g* for 10 min, after which the supernatant was discarded. Pellets were resuspended in 30 ml lysis buffer (25 mM Tris-HCl pH 8, 150 mM NaCl, 30 mM imidazole, 1 mM PMSF, 0.75% CHAPS, 1 mM DNase and 10 mM lysozyme, with Thermo Fisher Scientific Pierce protease inhibitor tablet). Cell suspensions were lysed by microfluidizer or sonication, and the lysate was clarified at 20,000*g* for around 30 min. The His-tagged proteins were bound to Ni-NTA resin (Qiagen) during gravity flow and washed with a wash buffer (25 mM Tris-HCl pH 8, 150 mM NaCl and 30 mM imidazole). Protein was eluted with an elution buffer (25 mM Tris-HCl pH 8, 150 mM NaCl and 300 mM imidazole). For the first-round designs, the His tag was removed by TEV cleavage, followed by IMAC purification to remove TEV protease. The flowthrough was collected and concentrated before further purification by SEC or fast-performance liquid chromatography

on a Superdex 200 increase 10/300 GL column in Tris-buffered saline (TBS; 25 mM Tris pH 8.0 and 150 mM NaCl).

## Circular dichroism

Circular dichroism spectra were measured with an AVIV Model 420 DC or Jasco J-1500 circular dichroism spectrometer. Samples were 0.25 mg ml$^{-1}$ in TBS (25 mM Tris pH 8.0 and 150 mM NaCl), and a 1-mm path-length cuvette was used. The circular dichroism signal was converted to mean residue ellipticity by dividing the raw spectra by $N \times C \times L \times 10$, in which $N$ is the number of residues, $C$ is the concentration of protein and $L$ is the path length (0.1 cm).

## SEC with multi-angle light scattering

Purified samples after the initial SEC run were pooled then concentrated or diluted as needed to a final concentration of 2 mg ml$^{-1}$ and 100 µl of each sample was then run through a high-performance liquid chromatography system (Agilent) using a Superdex 200 10/300 GL column. These fractionation runs were coupled to a multi-angle light scattering detector (Wyatt) to determine the absolute molecular weights for each designed protein as described previously[21].

## SAXS

SAXS was collected at the SIBYLS High Throughput SAXS Advanced Light Source in Berkeley, California[43,44]. Beam exposures of 0.3 s for 10.2 s resulted in 33 frames per sample. Data were collected at low (around 1.5 mg ml$^{-1}$) and high (around 2–3 mg ml$^{-1}$) protein concentrations in SAXS buffer (25 mM Tris pH 8.0, 150 mM NaCl and 2% glycerol). The SIBYLS website (SAXS FrameSlice) was used to analyse the data for high- and low-centration samples and average the best dataset. If there was obvious aggregation over the 33 frames, only the data points before aggregation arose were used in the Gunier region; otherwise, all data were included for the Gunier region. All data were used for the Porod and Wide regions. The averaged file was used with scatter.jar to remove data points with outlier residuals in the Gunier region. Finally, the data were truncated at 0.25 q. This dataset was then compared to the predicted SAXS profile based on the design model using the FoXS SAXS server (FoXS Server: Fast X-Ray Scattering n.d.), and the volatility ratio (Vr) was calculated to quantify how well the predicted data matched the experimental data. Proteins with a Vr of less than 2.5 were considered to be folded to the designed quaternary shape.

## Bio-layer interferometry

Bio-layer interferometry binding data were collected in an Octet RED96 (ForteBio) and processed using the instrument's integrated software. To measure the affinity of peptide binders, N-terminally biotinylated (biotin-Ahx) target peptides with a short linker (GGS) were loaded onto streptavidin-coated biosensors (SA ForteBio) at 50–100 nM in binding buffer (10 mM HEPES (pH 7.4), 150 mM NaCl, 3 mM EDTA, 0.05% surfactant P20 and 0.5% non-fat dry milk) for 120 s. Analyte proteins were diluted from concentrated stocks into the binding buffer. After baseline measurement in the binding buffer alone, the binding kinetics were monitored by dipping the biosensors in wells containing the target protein at the indicated concentration (association step) and then dipping the sensors back into baseline buffer (dissociation).

## Yeast surface display

*Saccharomyces cerevisiae* EBY100 strain cultures were grown in C-Trp-Ura medium and induced in SGCAA medium following the protocol in ref. 45. Cells were washed with PBSF (phosphate-buffered saline (PBS) with 1% BSA) and labelled with biotinylated designed proteins using two labelling methods: with-avidity and without-avidity labelling. For the with-avidity method, the cells were incubated with biotinylated RBD, together with anti-Myc fluorescein isothiocyanate (FITC, Miltenyi Biotec) and streptavidin–phycoerythrin (SAPE, Thermo Fisher Scientific). The SAPE in the with-avidity method was used at one-quarter of the concentration of the biotinylated RBD. The with-avidity method was used in the first few rounds of screening against the repeat-peptide library to fish out weak binder candidates. For the without-avidity method, the cells were first incubated with biotinylated designed proteins, washed and then secondarily labelled with SAPE and FITC.

## Crystallization and structure determination

**RPB_PEW3_R4–PAWx4.** Purified RPB_PEW3_R4 protein + PAWx4 peptide at a concentration of 36 mg ml$^{-1}$ was used to conduct sitting-drop, vapour-diffusion crystallization trials using the JCSG Core I-IV screens (NeXtal Biotechnologies). Crystals of RPB_PEW3_R4–PAWx4 grew from drops consisting of 100 nl protein plus 100 nl of a reservoir solution consisting of 0.1 M MES pH 5.0 and 30% (w/v) PEG 6000 at 4 °C, and were cryoprotected by supplementing the reservoir solution with 5% ethylene glycol. Native diffraction data were collected at APS beamline 23-ID-D, indexed to $P2_12_12_1$ and reduced using XDS[46] (Supplementary Table 1). The structure was phased by molecular replacement using Phaser[46]. A set of around 50 of the lowest-energy predicted models from Rosetta were used as search models. Several of these models gave clear solutions, which were adjusted in Coot[47] and refined using PHENIX[48]. Model refinement in $P2_12_12_1$ initially resulted in unacceptably high values for $R_{free} - R_{work}$. Refinement was therefore first performed in lower-symmetry space groups ($P1$ and $P2_1$). In the late stages of refinement, these $P1$ and $P2_1$ models were refined against the $P2_12_12_1$, which ultimately yielded acceptable, albeit somewhat higher, R-factors.

**RPB_PLP3_R6–PLPx6.** Purified RPB_PLP3_R6 protein + PLPx4 peptide at a concentration of 70 mg ml$^{-1}$ was used to conduct sitting-drop, vapour-diffusion crystallization trials using the JCSG Core I-IV screens (NeXtal Biotechnologies). Crystals of RPB_PLP3_R6-PLPx6 grew from drops consisting of 100 nl protein plus 100 nl of a reservoir solution consisting of 2.4 M $(NH_4)_2SO_4$ and 0.1 M sodium citrate pH 4 at 18 °C, and were cryoprotected by supplementing the reservoir solution with 2.2 M sodium malonate pH 4. Native diffraction data were collected at APS beamline 23-ID-D, indexed to $I422$ and reduced using XDS[49] (Supplementary Table 1). The structure was phased by molecular replacement using Phaser[46]. A set of around 28 of the lowest-energy predicted models from Rosetta were used as search models. Several of these models gave clear solutions, which were adjusted in Coot[47] and refined using PHENIX[48].

**RPB_LRP2_R4–LRPx4.** Purified RPB_LRP2_R4 protein + LRPx4 peptide at a concentration of 21.4 mg ml$^{-1}$ was used to conduct sitting-drop, vapour-diffusion crystallization trials using the JCSG Core I-IV screens (NeXtal Biotechnologies). Crystals of RPB_LRP2_R4–LRPx4 grew from drops consisting of 100 nl protein plus 100 nl of a reservoir solution consisting of 0.1 M HEPES pH 7 and 10% (w/v) PEG 6000 at 18 °C, and were cryoprotected by supplementing the reservoir solution with 25% ethylene glycol. Native diffraction data were collected at APS beamline 23-ID-B, indexed to $P32\,2\,1$ and reduced using XDS[49] (Supplementary Table 1). The structure was phased by molecular replacement using Phaser[46]. The coordinates of apo-RPB_LRP2_R4 from the proteolysed or filament structure were used as a search model. The resulting model was adjusted in Coot[47] and refined using PHENIX[48]. Like the apo structure, this crystal structure of RPB_LRP2_R4 also contained infinitely long filaments in the crystal, this time with peptide bound.

**RPB_PLP1_R6–PLPx6.** Purified RPB_PLP1_R6 protein + PLPx6 peptide at a concentration of 143 mg ml$^{-1}$ was used to conduct sitting-drop, vapour-diffusion crystallization trials using the JCSG Core I-IV screens (NeXtal Biotechnologies). Crystals of RPB_PLP1_R6–PLPx6 grew from drops consisting of 100 nl protein plus 100 nl of a reservoir solution consisting of 0.2 M NaCl and 20% (w/v) PEG 3350 at 4 °C, and were cryoprotected by supplementing the reservoir solution with 15% ethylene glycol. Native diffraction data were collected at APS beamline 23-ID-B,

indexed to H32 and reduced using XDS[49] (Supplementary Table 1). The structure was phased by molecular replacement using Phaser[46]. A set of around 230 of the lowest-energy predicted models from Rosetta were used as search models. Several of these models gave clear solutions, which were adjusted in Coot[47] and refined using PHENIX[48]. In the later stages of refinement, two copies of the 6xPLP peptide were built into clearly defined electron density in the asymmetrical unit. The first copy adopts the expected location based on the design, and makes the designed interactions with RPB_PLP1_R6. The density for this peptide and the final atomic model (19 amino acid residues) are slightly longer than the peptide used in crystallization (18 residues); this is probably due to 'slippage' or misregistration of the peptide relative to the R6PO11 in many unit cells, resulting in density longer than the peptide itself. A second copy of the peptide lies across a twofold symmetry axis at around 50% occupancy, resulting in the superposition of this peptide with a symmetry-derived copy of itself running in the opposite direction. Despite this, the locations of each Pro or Leu side-chain unit were reasonably well defined. However, it seems unlikely that the binding of the peptide at this second site would occur readily in solution.

**RPB_PLP1_R6, alternative conformation 1.** Purified RPB_PLP1_R6 protein + PLPx6 peptide at a concentration of 166 mg ml$^{-1}$ was used to conduct sitting-drop, vapour-diffusion crystallization trials using the JCSG Core I-IV screens (NeXtal Biotechnologies). Crystals of RPB_PLP1_R6-PLPx6 grew from drops consisting of 100 nl protein plus 100 nl of a reservoir solution consisting of 0.02 M CaCl$_2$, 30% (v/v) MPD and 0.1 M sodium acetate pH 4.6 at 18 °C, and were cryoprotected by supplementing the reservoir solution with 5% MPD. Native diffraction data were collected at APS beamline 23-ID-B, indexed to P22121 and reduced using XDS[49] (Supplementary Table 1). The structure was phased by molecular replacement using Phaser[46], using the coordinates for R6PO11 (alternative conformation 1) as a search model. The model was adjusted in Coot[47] and refined using PHENIX[48]. In the later stages of refinement, one copy of the 6xPLP peptide was model at a site of crystal contact, where it is sandwiched between adjacent subunits in a way that is likely to only be bound in the crystal lattice.

**RPB_PLP1_R6, alternative conformation 2.** Purified RPB_PLP1_R6 protein + PLPx6 peptide at a concentration of 166 mg ml$^{-1}$ was used to conduct sitting-drop, vapour-diffusion crystallization trials using the JCSG Core I-IV screens (NeXtal Biotechnologies). Crystals of RPB_PLP1_R6-PLPx6 grew from drops consisting of 100 nl protein plus 100 nl of a reservoir solution consisting of 40% (v/v) MPD and 0.1 M sodium phosphate-citrate pH 4.2 at 18 °C, and were cryoprotected by supplementing the reservoir solution. Native diffraction data were collected at APS beamline 23-ID-B, indexed to P22121 and reduced using XDS[49] (Supplementary Table 1). Initial attempts to phase by molecular replacement using Phaser[46] and around 500 predicted models from Rosetta and RoseTTAfold failed to yield any clear solutions. Similarly, several thousand truncations of these models (containing all combinations of 1, 2, 3, 4 or 5 of the 6 repeat units) also failed to give clear solutions. To try to identify correct but low-scoring solutions in the output of these trials, we ran SHELXE autobuilding and density modification on a large number of these potential solutions. Ultimately, we were able to identify an MR solution with two out of six repeats correctly placed that allowed the autobuilding of a polyalanine model and an interpretable map, which could be further improved by iterative rounds of rebuilding in Coot[47] and refinement using PHENIX[48]. Ultimately, the final model revealed that in this crystal form and a similar crystallization condition (RPB_PLP1_R6, alternative conformation 1, above), RPB_PLP1_R6 adopted an alternative fold.

**RPB_LRP2_R4.** Purified RPB_LRP2_R4−LRPx4 protein at a concentration of 33 mg ml$^{-1}$ was used to conduct sitting-drop, vapour-diffusion crystallization trials using the JCSG Core I-IV screens (NeXtal Biotechnologies). Crystals of RPB_LRP2_R4 grew from drops consisting of 100 nl protein plus 100 nl of a reservoir solution consisting of 0.2 M K$_2$HPO$_4$ and 20% (w/v) PEG 3350 at 18 °C, and were cryoprotected by supplementing the reservoir solution with 15% ethylene glycol. Native diffraction data were collected at APS beamline 23-ID-B, indexed to P32 2 1 and reduced using XDS[49] (Supplementary Table 1). The structure was phased by molecular replacement using Phaser[46]. A set of around 50 of the lowest-energy predicted models from Rosetta, as well as a variety of truncated models, were used as search models. Several of these models gave clear solutions, which were adjusted in Coot[47] and refined using PHENIX[48]. Four helical-repeat modules were present in the asymmetrical unit. However, unexpectedly, side-chain densities for all four repeats were very similar to one another and matched the sequence of the internal helical repeats, but not the N- and C-terminal capping repeats, which are slightly different from the internal ones. In addition, these four repeat units pack tightly against adjacent, symmetry-related molecules such that they form an 'infinitely long' repeat protein running throughout the crystal. Careful examination of the the junction between each repeat unit revealed no clear breaks in electron density; the density for the backbone is continuous through the asymmetrical unit, and continuous with the symmetry-related molecules near the N terminus and C terminus of the molecule in the asymmetrical unit. Rather than truly forming an infinitely long polymer, we suspect that proteolytic cleavage of the RPB_LRP2_R4 (either during purification or crystallization) led to the removal of the N- and/or C-terminal caps in many molecules, which could allow the internal repeats from separate molecules to polymerize to form fibres in the crystal. Heterogeneity in these cleavage products and how they assemble into the crystal lattice (misregistration) could consequently explain the 'continuous' filaments of this repeat protein that we observe in these crystals.

## Cell studies

**Plasmids.** For expression in cells, constructs were synthesized by Genescript and cloned into a modified pUC57 plasmid (GenScript) allowing mammalian expression under a EF1a promoter. Target peptides were cloned as C-terminal fusions with a linker (GAGAGAGRP) followed by EGFP. Binders were expressed as fusions with an N-terminal Mito-Tag−the first 34 residues of the Mas70p protein, shown to efficiently relocalize proteins to mitochondria in mammalian cells[50]−and a C-terminal mScarlet tag[51]. Plasmids encoding the GFP-tagged peptide and the mScarlet-tagged binder were then cotransfected into cells.

Alternatively, for an in vivo demonstration of the multiplexed binding between different peptides and their cognate binders (Fig. 3f,g), bicistronic plasmids were generated expressing the binder flanked with a Mito-Tag followed by a stop codon, then an internal ribosome entry site (IRES) sequence and the target peptide tagged with EGFP. Alternatively, the binder was flanked with a PEX tag−the first 66 residues of human PEX3, targeting to peroxisomes[52]−and the target peptide was tagged with mScarlet. Cells were then cotransfected with both bicistronic plasmids to express all four proteins.

**Cells.** U2OS FlipIn Trex cells (a gift from S. C. Blacklow) and HeLa FlpIn Trex cells (a gift from S. Bullock), were cultured in DMEM (Corning) supplemented with 10% fetal bovine serum (Gibco) and 1% penicillin–streptomycin (Gibco) at 37 °C with 5% CO$_2$. Cells were transfected with Lipofectamine 3000 (Invitrogen) according to the manufacturer's instructions, and imaged after one day of expression. Cell lines were not authenticated. Cells were routinely screened for mycoplasma by DAPI staining.

**Live-cell imaging.** For live-cell imaging (Fig. 3), U2OS FlipIn Trex cells were plated on glass-bottom dishes (World Precision Instruments, FD35) coated with fibronectin (Sigma, F1141, 50 μg ml$^{-1}$ in PBS), for 1 h at 37 °C in DMEM-10% serum. Medium was then changed to Leibovitz's L-15 medium (Gibco) supplemented with 20 mM HEPES (Gibco) for live-cell

imaging. Imaging was performed using a custom spinning disk confocal instrument composed of a Nikon Ti stand equipped with a perfect focus system, a fast Z piezo stage (ASI) and a PLAN Apo Lambda 1.45 NA 100× objective, and a spinning disk head (Yokogawa CSUX1). Images were recorded with a Photometrics Prime 95B back-illuminated sCMOS camera run in pseudo global shutter mode and synchronized with the spinning disk wheel. Excitation was provided by 488 and 561 lasers (Coherent OBIS mounted in a Cairn laser launch) and imaged using dedicated single-bandpass filters for each channel mounted on a Cairn Optospin wheel (Chroma 525/50 for GFP and Chroma 595/50 for mScarlet). To enable fast 4D acquisitions, an FPGA module (National Instrument sbRIO-9637 running custom code) was used for hardware-based synchronization of the instrument, in particular to ensure that the piezo z stage moved only during the readout period of the sCMOS camera. The temperature was kept at 37 °C using a temperature control chamber (MicroscopeHeaters.Com). The system was operated by Metamorph.

**Immunofluorescence.** For immunofluorescence of mitochondria (Extended Data Fig. 2b), U2OS FlpIn Trex cells (a gift from S. C. Blacklow) were spread on glass-bottom dishes coated with fibronectin as above. Cells were washed with PBS then fixed in 4% PFA for 20 min at room temperature. After fixation, cells were washed with PBS and then permeabilized with 0.1% Triton X-100 in PBS for 5 min at room temperature. Cells were washed again with PBS and blocked in 1% BSA in PBS for 15 min. Cells were then incubated with TOM20 antibody (Santa Cruz, sc-17764, used at 1:200 dilution), diluted in 1% BSA in PBS, for 1 h at room temperature. Cells were washed three times with PBS and then incubated with DAPI (Roche, 10236276001) and anti-mouse Alexa Fluor 488, diluted at 1:400 in 1% BSA in PBS, for 1 h at room temperature. Cells were washed a final three times in PBS and then imaged using the spinning disk confocal described above.

**Pull-down of endogenous proteins from extracts using designed binders.** For the pull-down of endogenous ZFC3H1 from human cell extracts, HeLa FlpIn Trex cells were lysed in lysis buffer (25 mM HEPES, 150 mM NaCl, 0.5% Tx100, 0.5% NP-40 and 20 mM imidazole, pH 7.4, supplemented with Roche EDTA-free protease inhibitor tablets). The lysate was incubated on ice for 10 min to continue lysis and then spun at 4,000g for 15 min at 4 °C. The supernatant was incubated with pre-washed Ni-NTA agarose (Qiagen, 30210 318/AV/01) for 1 h with rocking at 4 °C to remove or reduce proteins in the lysate that bind to the resin non-specifically. For each condition, 50 μl of fresh Ni-NTA agarose resin was washed twice in lysis buffer. Equimolar amounts of purified His-tagged binder, or as a control an equal volume of buffer, was added to the Ni-NTA agarose. The pre-cleared HeLa lysate was split evenly between the three conditions. An input was taken of each condition, and the tubes were incubated for 2 h at 4 °C with rocking. Beads were then washed twice in lysis buffer and twice in wash buffer (25 mM HEPES, 150 mM NaCl and 20 mM imidazole pH 7.4). Proteins were then eluted from the beads in elution buffer (25 mM HEPES, 150 mM NaCl and 500 mM imidazole, pH 7.4). Inputs and elutions were run on a NuPage 3-8% Tris-Acetate gel (Invitrogen, EA0375) and transferred to a nitrocellulose membrane using the iBlot system (Thermo Fisher Scientific). Membranes were blocked in 5% (w/v) milk in TBS-TWEEN (10 mM Tris-HCl, 120 mM NaCl and 1% (w/v) TWEEN20, pH 7.4) for 30 min at room temperature with gentle shaking. Rabbit anti-ZFC3H1 (Sigma, HPA007151, used at 1:250) and mouse anti-α-tubulin 488 (Clone DMA1, Sigma T6199, directly labelled with Abberior STAR 488, NHS ester leading to a 4.5 dye/antibody degree of labelling, and used at 0.1 μg ml⁻¹ final concentration) were diluted in 1% (w/v) milk in TBS-TWEEN and incubated with the membrane overnight at 4 °C with gentle shaking. The membrane was washed three times in TBS-TWEEN then incubated with goat anti-rabbit Alexa 555 (Invitrogen, A32732, 1:2,000) for 1 h at room temperature with gentle shaking. The membrane was washed twice with TBS-TWEEN, followed by a final wash with TBS-TWEEN

with 0.001% SDS. Membranes were imaged using a ChemiDoc system (BioRad). Alternatively, the same samples were analysed using 4–12% Bis-Tris gels (Invitrogen NP0323BOX) and stained with InstantBlue Coomassie stain (Sigma ISB1L). Note that αZFC-high was also able to pull down endogenous ZFC3H1 from human cell extracts when 50 mM rather than 150 mM NaCl was used in all buffers (Extended Data Fig. 7b).

**Mass spectrometry.** Each line of the polyacrylamide gel presented in Fig. 6c was cut into six pieces (1–2 mm) and prepared for mass spectrometric analysis by manual in situ enzymatic digestion (the gel area containing the binder was omitted from the analysis to avoid saturation of the detector by overabundance of binder peptides). In brief, the excised protein gel pieces were placed in a well of a 96-well microtitre plate and destained with 50% (v/v) acetonitrile and 50 mM ammonium bicarbonate, reduced with 10 mM DTT and alkylated with 55 mM iodoacetamide. After alkylation, proteins were digested with 6 ng μl⁻¹ trypsin (Promega) and 0.1% Protease Max (Promega) overnight at 37 °C. The resulting gel pieces were extracted with ammonium bicarbonate (100 μl, 100 mM) and ammonium bicarbonate/acetonitrile (50/50, 100 μl) before being dried down by vacuum. Clean-up of peptide digests was carried out with HyperSep SpinTip P-20 (Thermo Fisher Scientific) C18 columns, using 80% acetonitrile as the elution solvent before being dried down again. The resulting peptides were extracted in 0.1% (v/v) trifluoroacetic acid acid and 2% (v/v) acetonitrile. The digest was analysed by nano-scale capillary liquid chromatography–tandem mass spectrometry (LC–MS/MS) using an Ultimate U3000 HPLC (Dionex, Thermo Fisher Scientific) to deliver a flow of 250 nl min⁻¹. Peptides were trapped on a C18 Acclaim PepMap100 5 μm, 100 μm × 20 mm nanoViper (Thermo Fisher Scientific) before separation on a PepMap RSLC C18, 2 μm, 100 A, 75 μm × 75 cm EasySpray column (Thermo Fisher Scientific). Peptides were eluted on a 90-min gradient with acetonitrile and interfaced using an EasySpray ionization source to a quadrupole Orbitrap mass spectrometer (Q-Exactive HFX, Thermo Fisher Scientific). Mass spectrometry data were acquired in data-dependent mode with a top-25 method; high-resolution full mass scans were performed ($R = 120,000$, $m/z$ 350–1,750), followed by higher-energy collision dissociation with a normalized collision energy of 27%. The corresponding tandem mass spectra were recorded ($R = 30,000$, isolation window $m/z$ 1.6, dynamic exclusion 50 s). LC–MS/MS data were then searched against the Uniprot human proteome database, using the Mascot search engine programme (Matrix Science)[53]. Database search parameters were set with a precursor tolerance of 10 ppm and a fragment ion mass tolerance of 0.1 Da. One missed enzyme cleavage was allowed and variable modifications for oxidation, carboxymethylation and phosphorylation. MS/MS data were validated using the Scaffold programme (Proteome Software)[54]. All data were in addition interrogated manually. To generate the Venn diagram in Fig. 6f, we considered a threshold of minimum five peptides to consider that a protein had been identified. The mass spectrometry proteomics data have been deposited to the ProteomeXchange Consortium through the PRIDE[55] partner repository with the dataset identifiers PXD038492 and 10.6019/PXD038492. See also Source Data for the annotated full dataset.

## Reporting summary

Further information on research design is available in the Nature Portfolio Reporting Summary linked to this article.

## Data availability

The atomic coordinates and experimental data of RPB_PEW3_R4–PAWx4, RPB_PLP3_R6–PLPx6, RPB_LRP2_R4–LRPx4, RPB_PLP1_R6–PLPx6, RPB_PLP1_R6–PLPx6 (alternative conformation 1), RPB_PLP1_R6–PLPx6 (alternative conformation 2) and RPB_LRP2_R4 (pseudopolymeric) have been deposited in the RCSB PDB with the accession

numbers 7UDJ, 7UE2, 7UDK, 7UDL, 7UDM, 7UDN and 7UDO, respectively. The Rosetta macromolecular modelling suite (https://www.rosettacommons.org) is freely available to academic and non-commercial users. Commercial licences for the suite are available through the University of Washington Technology Transfer Office. The mass spectrometry proteomics data have been deposited to the ProteomeXchange Consortium through the PRIDE partner repository with the dataset identifiers PXD038492 and 10.6019/PXD038492. Source data are provided with this paper. All protein sequences for the binders described in this study are provided in Supplementary Table 2.

## Code availability

The design scripts and main PDB models, computational protocol for data analysis, experimental data and analysis scripts, all the design models and the next-generation-sequencing results used in this paper can be downloaded from file servers hosted by the Institute for Protein Design: https://files.ipd.uw.edu/pub/2023_modular_peptide_binding_proteins/all_data_modular_peptide_binding_proteins.tar.gz. The code to identify proteins in databases containing any linear combination of amino acid triplets given as an input can be found on GitHub (https://github.com/tjs23/prot_pep_scan).

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

**Acknowledgements** We thank B. Wicky, A. Ljubetic and I. Lutz for advice on the split luciferase assay for the second-round design screening; C. Xu for help troubleshooting experiments; T. Schlichtharle for discussion; L. Cao for advice on bio-layer interferometry; H. Pyles for advice on circular dichroism and DHR proteins; R. Hegde for the suggestion to target disordered regions of endogenous proteins; and K. Van Wormer and A. Curtis Smith for laboratory support during COVID-19. This work was supported by the Audacious Project at the Institute for Protein Design (D.B., K.W., M.D., D.A.S. and A.B.); the Michelson Found Animals Foundation grant number GM15-S01 (L.S., K.W. and D.B.); the National Institute on Aging grant 5U19AG065156-02 (D.R.H., K.W. and D.B.); the National Institute of General Medical Sciences grant R35GM128777 (D.C.E.); the Howard Hughes Medical Institute (D.B., W.S. and H.B.); the Open Philanthropy Project Improving Protein Design Fund (Y.-T.C., R.R., C.M.C., G.B., D.C.E. and D.B.); the Donald and Jo Anne Petersen Endowment for Accelerating Advancements in Alzheimer's Disease Research (T.J.B. and D.B.); a donation from AMGEN to the Institute for Protein Design (I.G.); the Medical Research Council (MC_UP_1201/13 to E.D., T.E.M. and T.J.S.); the Human Frontier Science Program (CDA00034/2017-C to E.D.); and a Sir Henry Wellcome Postdoctoral Fellowship (220480/Z/20/Z to K.E.M.).

**Author contributions** K.W., D.A.S. and D.B. designed the research. D.A.S. and D.B. developed the preliminary computational method and hash database. W.S. contributed to the development of the hash database. K.W. updated the computational method with help from D.A.S. and H.B. H.B. updated the hash database to be more general. Y.S. helped and contributed to the first development of the hash database. K.W. designed the polyproline II DHR scaffold library using the method developed by D.R.H. K.W. designed the binders with help from H.B. H.B. and K.W. performed the yeast screening, expression and binding experiments with help from I.G. for the first-round design characterization. K.W. performed bio-layer interferometry and Octet assays for the second-round design characterization. H.B. constructed and screened SSM libraries. Y.-T.C., R.R., G.B. and D.C.E. solved the structures of RPB_PEW3_R4–PEWx4, RPB_PLP3_R6–PLPx6, RPB_LRP2_R4–LRPx4 and RPB_PLP1_R6–PLPx6. K.E.M. designed and performed all cell experiments in this work, in particular the multiplex binding assay and the demonstration of the endogenous binder for ZFC3H1. E.D. identified ZFC3H1 as a good target for the development of an endogenous binder with help from T.J.S. T.E.M. performed mass spectrometry analysis. A.B. helped with the modular binding assay. M.D. and C.M.C. helped with preparing protein samples for crystallography. All authors analysed data. L.S., D.A.S. and D.B. supervised research. K.W. and D.B. wrote the manuscript with input from the other authors. All authors revised the manuscript.

**Competing interests** The authors declare no competing interests, except as follows. K.W., H.B., D.R.H., T.J.B., K.E.M., T.J.S., T.E.M., Y.-T.C., R.R., G.B., D.C.E., L.S., E.D., D.A.S., W.S., I.G. and D.B. are co-inventors on a patent application entitled 'De novo designed modular peptide binding proteins by superhelical matching' (63/381,109, filed 26 October 2022).

**Additional information**
**Correspondence and requests for materials** should be addressed to Emmanuel Derivery, Daniel Adriano Silva or David Baker.

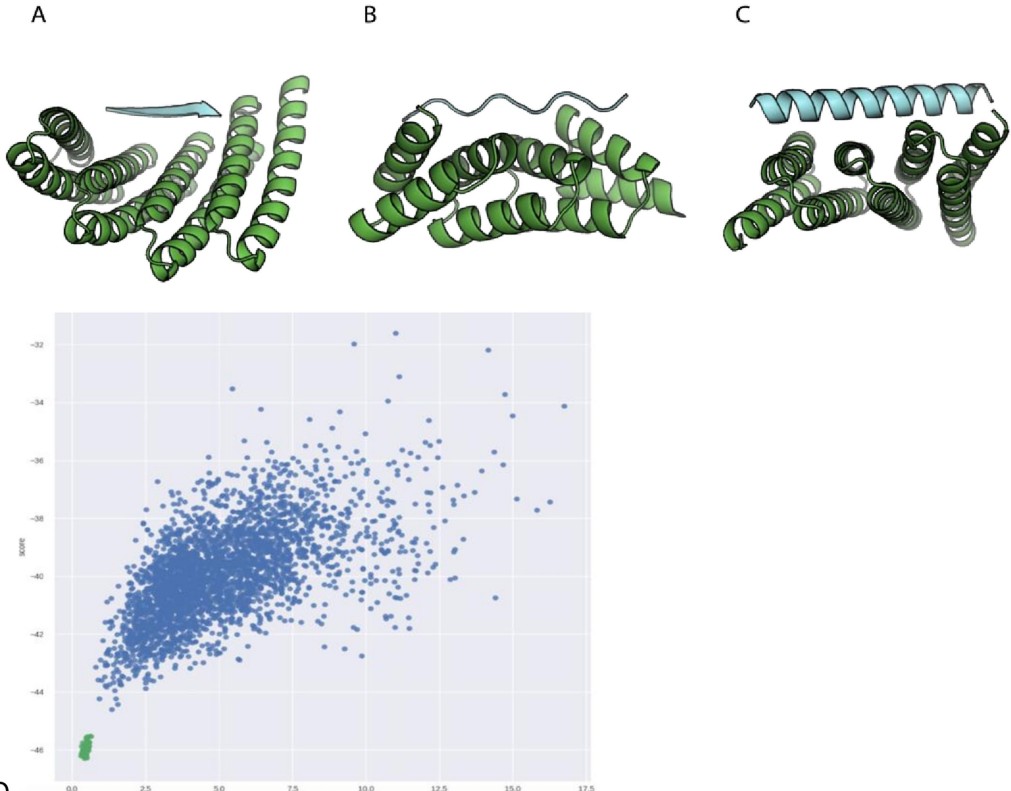

**Extended Data Fig. 1 | Examples of computationally designed model geometry and convergence of backbone docking. a–c**, Examples of repeat proteins computationally designed to bind to extended beta strand (**a**), polypeptide II (**b**) and helical peptide backbones (**c**). **d**, Monte Carlo flexible backbone docking calculations after design to assess the structural specificity of the designed peptide-binding interface. It started from large numbers of peptide conformations randomly generated with superhelical parameters in the range of those of the proteins (usually 10,000–50,000 trajectories), and selected those designs with converged peptide backbones (RMSD < 2.0 among the top 20 designs with lowest DDG) close to the design model (RMSD < 1.5). Green dots shown in the above example plot represent the converged designs picked by this threshold.

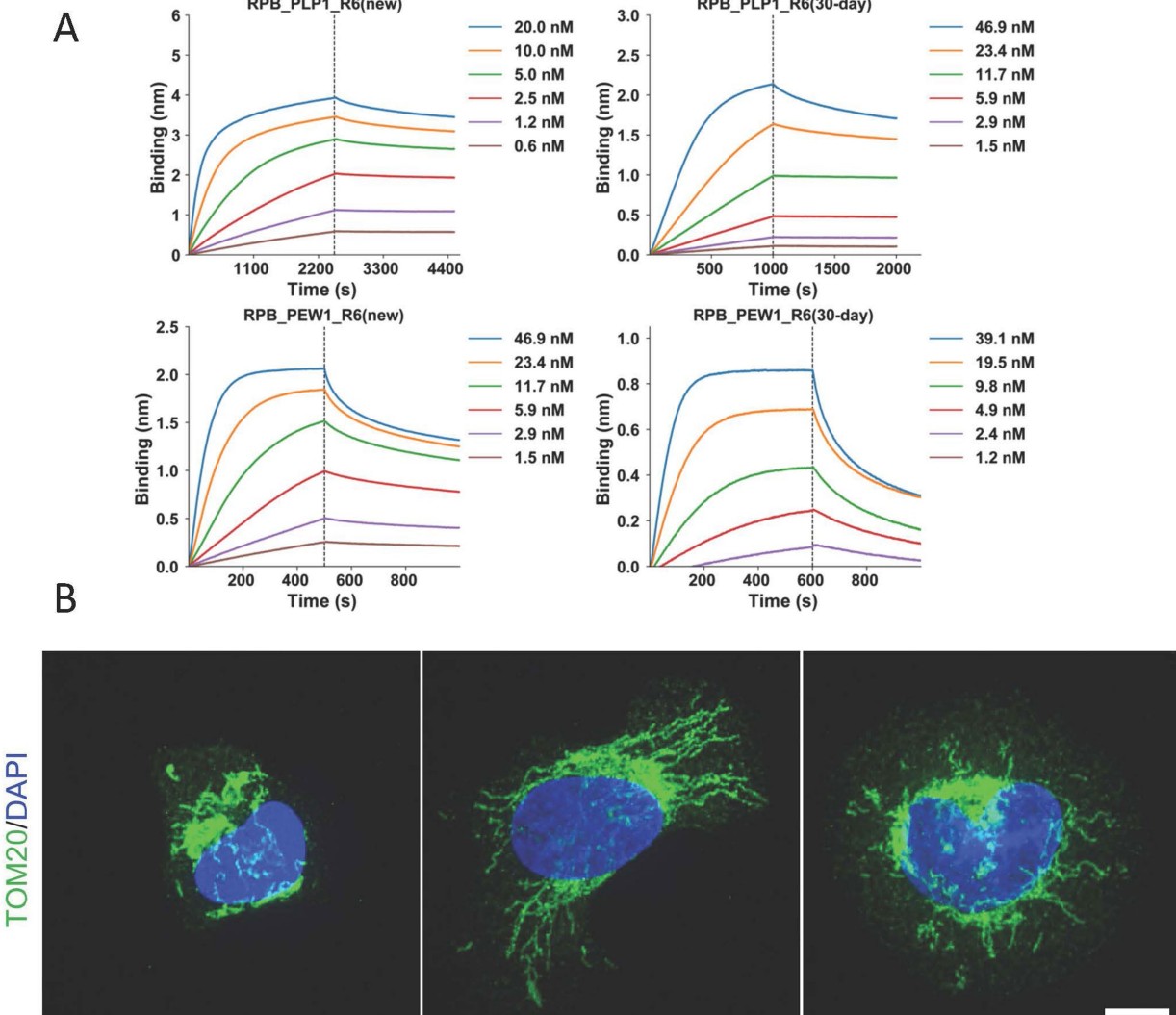

**Extended Data Fig. 2 | Comparison of binding affinities from freshly made and 30-day-old samples, and mitochondria immunostainings in control U2OS cells. a**, Little decrease in binding observed for designs RPB_PLP1_R6 and RPB_PEW1_R6 30-day-old in 4 °C. Bio-layer interferometry characterization of binding of designed proteins to the corresponding peptide targets. Twofold serial dilutions were tested for each binder, and the full tested concentration is labelled. The biotinylated target peptides were loaded onto the streptavidin (SA) biosensors, and incubated with designed binders in solution to measure association and dissociation. **b**, Mitochondria immunostainings in control U2OS cells. Wild-type U2OS cells were spread onto fibronectin coverslips as in Fig. 3, then fixed and processed for immunofluorescence using TOM20 antibodies as a marker of mitochondria. Note that mitochondria appearance in these control cells is similar to that observed upon overexpression of designed binders fused to mitochondria-targeting sequences (Fig. 3). suggesting that these constructs do not affect mitochondria shape. Scale bar, 10 μm.

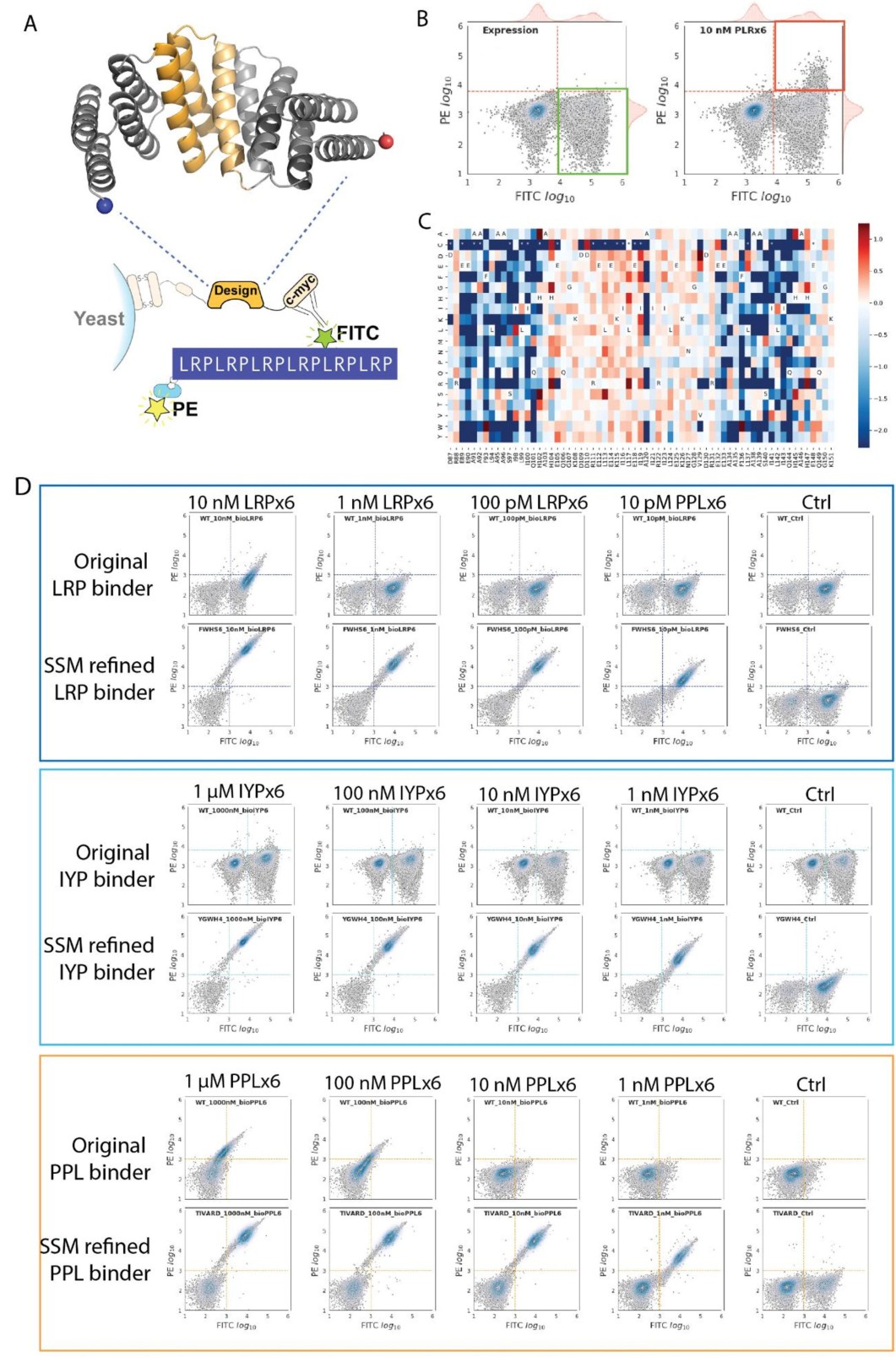

**Extended Data Fig. 3** | See next page for caption.

**Extended Data Fig. 3 | SSMs libraries are constructed and screened for enhancing the peptide-binding abilities of designed repeat-peptide binders. a**, A schematic illustration of the mutagenesis region within the designed repeat protein, and the principles of the yeast surface display assay for peptide binding analysis. In short, the biotinylated repeat peptides (a six-repeat of LRP peptide is shown as an example) are synthesized and can be detected by SAPE, while the expression of designed protein on yeast surface are monitored by FITC-conjugated anti-Myc antibody. A double high signal of both PE and FITC, using flow cytometry, indicates the valid peptide-binding events. **b**, The SSM libraries are first subjected to expression sorting (left), in which there is no targeted peptide added. The yeast populations, which display well expressed SSM mutants, will show above threshold FITC signals, are collected (green box) for next-generation sequencing, and are regrown for the next rounds of sorting. In the next round sorting, the targeted peptide is incubated with the yeast library, and labelled by both FITC and SAPE (right). The FITC⁺PE⁺ population is collected for analysis (orange box). **c**, By using next-generation sequencing, enrichment analysis for each mutation is carried out, and a heat map for all mutations is generated. In this heat map, using a designed LRP binder SSM library as an example, the red shades indicate enrichment with incubating with the targeted peptide, and the blue shades indicate depletion. Several mutations show exceptional enhancement of the LRP repeat peptide-binding ability, such as F93W, H102S and others. **d**, Using the SSM library, we can markedly enhance the peptide-binding abilities of the designed peptide binder. Three example yeast display assays titrating the peptide concentrations are shown here. The top row of each example is using the originally designed peptide binder, and the bottom row is using the peptide binder containing the combinations of the best mutations discovered in the SSM library screenings. An approximately 1,000-fold increase of the peptide-binding ability can be achieved with the assistance of SSM libraries. Note, the ratio of yeast population in the upper right quadrant indicates the peptide-binding ability.

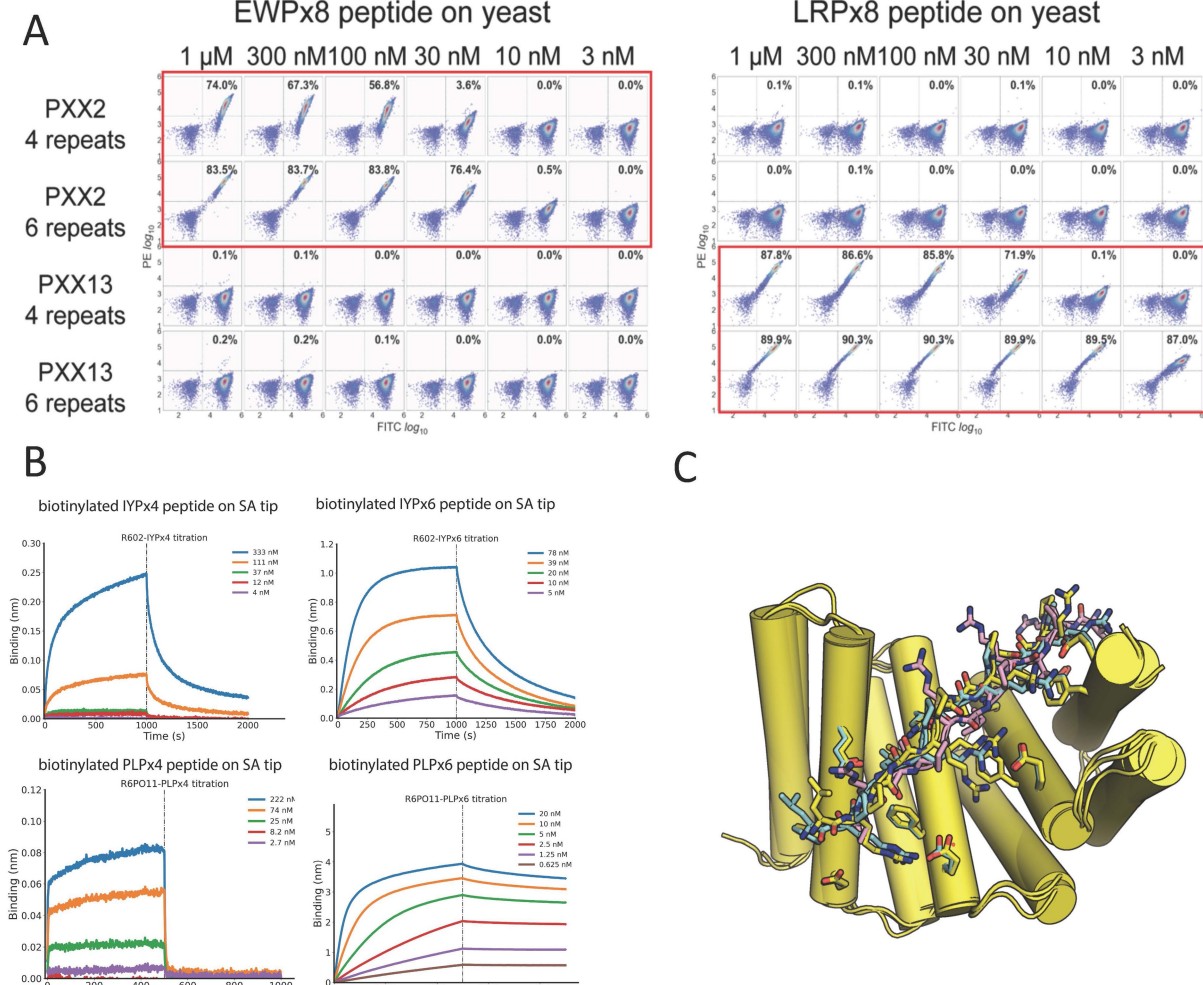

**Extended Data Fig. 4 | Comparison of binding affinities when changing repeat numbers from either binder or peptide side. and top five flexible backbone docks for the four-repeat LRP binder RPB_LRP2_R4–LRPx4. a**, Six-repeat versions of RPB_LRP2_R6 and RPB_PEW2_R6 had higher affinity for eight-repeat LRP and PEW peptides than four-repeat versions without any decrease in specificity in yeast surface display. Biotinylated repeat proteins (the six-repeat versions RPB_LRP2_R6 and RPB_PEW2_R6 and the four-repeat versions RPB_LRP2_R4 and RPB_PEW2_R4) were detected by SAPE, and the expression of the designed repeat peptide on yeast surface was monitored by FITC-conjugated anti-Myc antibody. Serial dilutions were tested for each binder, and the full tested concentration is labelled. **b**, Six-repeat IYP and PLP peptides had higher affinity for six-repeat versions of the cognate designed

repeat proteins (RPB_IYP1_R6 and RPB_PLP1_R6) than four-repeat versions by bio-layer interferometry. The full tested concentration is labelled. The biotinylated target peptides were loaded onto the streptavidin (SA) biosensors, and incubated with designed binders in solution to measure association and dissociation. The dissociation rate was markedly increased when testing against the six-repeat peptides as compared to the four-repeat peptides, indicating a much tighter binding event. **c**, Top five complex PDBs for RPB_LRP2_R4–LRPx4 from the flexible docking generated ensemble. Green, pink and grey are the ones closest to the crystal structure (shown in yellow) with RMSD over the peptide and the binding residues ≈ 0.03 Å, whereas the cyan dock RMSD = 3.89 Å.

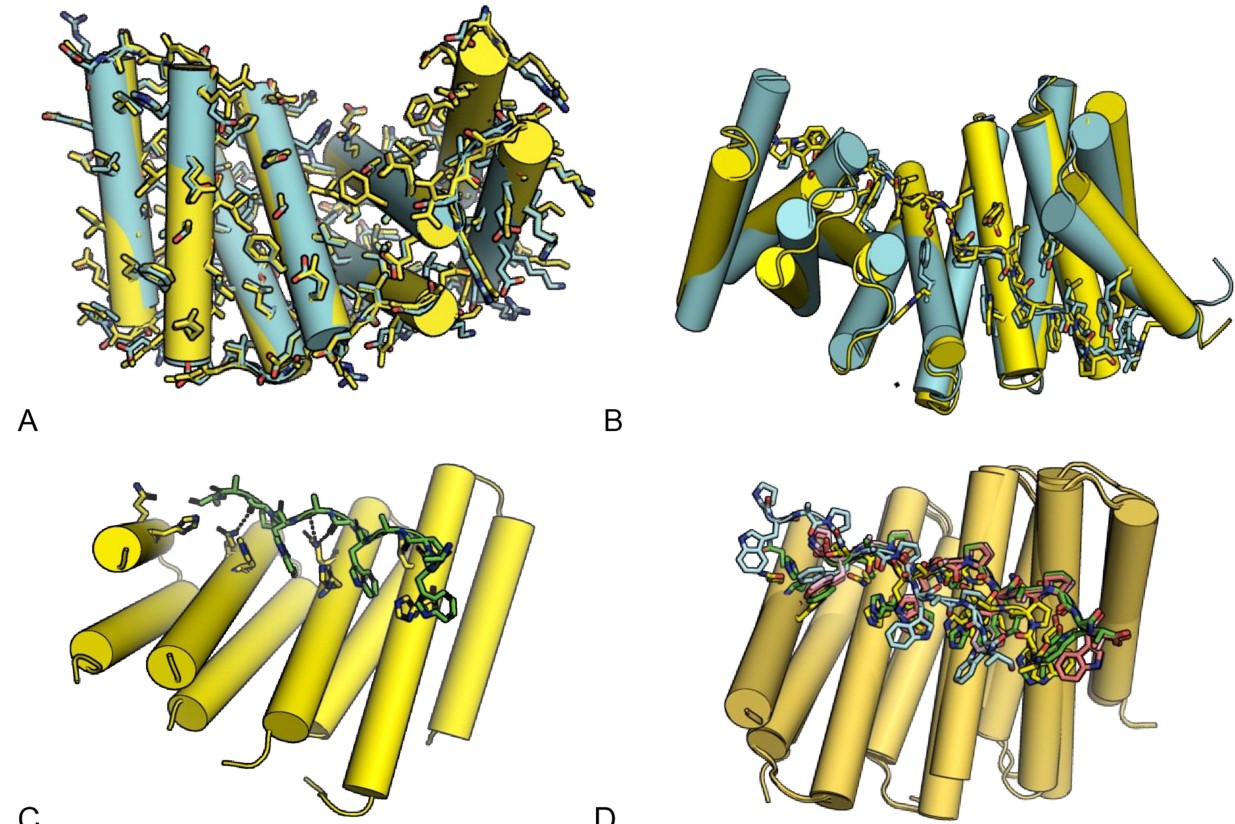

A

B

C

D

**Extended Data Fig. 5 | Crystal structures of the unbound RPB_LRP2_R4, bound RPB_PLP3_R6–PLPx6 and bound RPB_PEW3_R4 and its top five flexible backbone docks. a**, Crystal structure of the unbound first-round design RPB_LRP2_R4 (yellow) aligned with the design model (cyan). **b**, Crystal structure of the first-round complex RPB_PLP3_R6–PLPx6 (yellow) aligned with the design model (cyan). As is shown here, the peptide PLP units fit exactly into the designed curved groove formed by repeating tyrosine, alanine and tryptophan residues matching the design model with near atomic accuracy, with Cα RMSD of 1.70 Å for the binder apo, 2.00 Å for the peptide neighbour interface and 1.64 Å for the whole complex. **c**, Co-crystal structure of RPB_PEW3_R4–PAWx4. The PAW units bind to a relatively flat groove formed by repeating histidine residues and glutamine residues as designed (shown as sticks). **d**, Top five complex PDBs for RPB_PEW3_R4–PAWx4 from the flexible docking generated ensemble. Green, pink and grey are the ones closest to the crystal structure (shown in yellow) with RMSD over the peptide and the binding residues ≈ 0.03 Å, whereas the cyan dock RMSD = 3.89 Å.

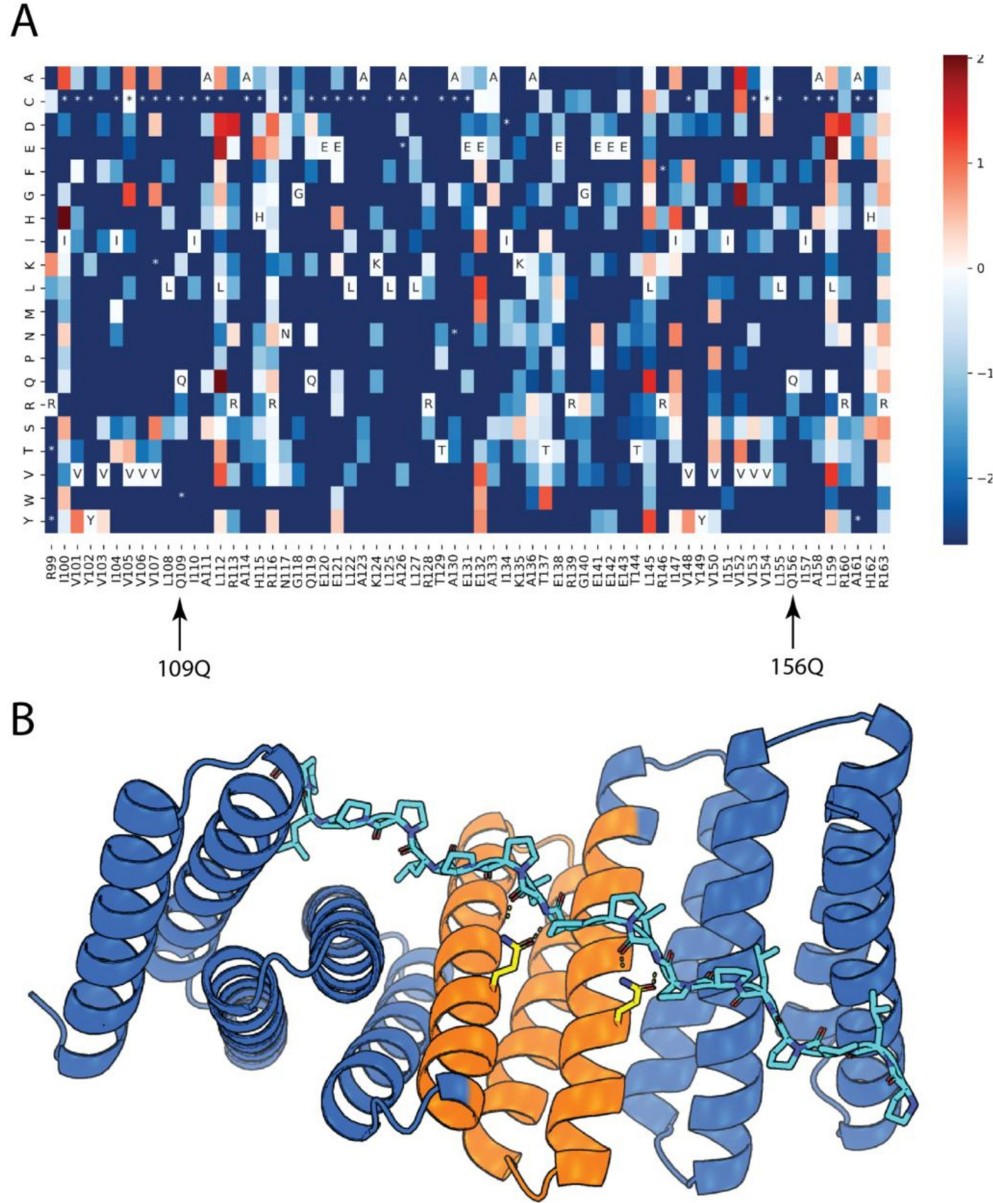

**Extended Data Fig. 6 | SSM binding interface footprinting results were consistent with the design model and crystal structure. a**, Using a PPL repeat-peptide binder as an example, a heat map presenting enrichment analysis for each mutation is generated. In each cell, the red colour indicates enrichment, and the blue colour indicates depletion. Wild-type sequences are indicated in the cells labelled with amino-acid one-letter codes. The mutants missing in the expression library are labelled with asterisks. Two positions (109Q and 156Q) are highlighted as examples showing conserved positions. Almost all mutations other than the wild type in these two positions are greatly depleted. **b**, Illustration shows the SSM region (orange), and the two conserved positions (109Q and 156Q in yellow).

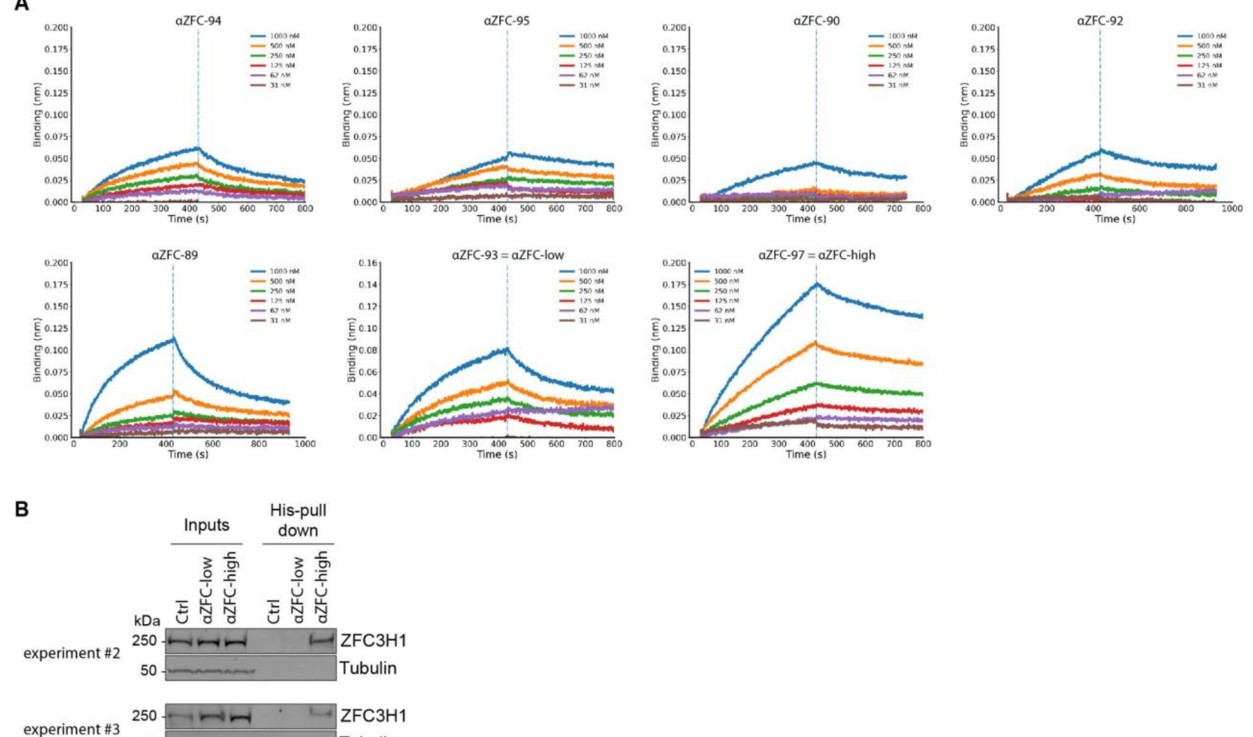

**Extended Data Fig. 7 | Characterization of ZFC3H1 binders. a**, Bio-layer interferometry screening for the seven endogenous ZFC3H1 binders. Twofold serial dilutions were tested for each binder, and the full tested concentration is labelled. The biotinylated target 24-amino-acid peptides (PLPPLPPLPPLPPEDP EQPPKPPF) were loaded onto the streptavidin (SA) biosensors, and incubated with designed binders in solution to measure association and dissociation. The two tightest binders (αZFC_93 and αZFC_97, renamed αZFC-high and αZFC-low, respectively) were selected for further fluorescence polarization characterization and cell assays. **b**, Characterization of ZFC3H1 binders for pull-down of endogenous target: Hela cell extracts were subjected to pull-down using the indicated binders bound to Ni-NTA agarose beads, or naked beads as control. Recovered proteins were processed for western blot against endogenous ZFC3H1 (or tubulin as a loading control). Two completely independent experiments are shown. These experiments are repeats of the experiment presented in Fig. 6e, albeit at a different salt concentration, namely 50 mM instead of 150 mM. For gel source data, see Supplementary Fig. 1.

**Extended Data Table 1 | Summaries of first- and second-round experimental characterization**

A

| Design ID | Design target | Experimental target | di-peptide target | Tri-peptide target |
|---|---|---|---|---|
| PXX02 | PSD | PEW | | 1 |
| PXX10 | PMP | PWP/PAW | | 2 |
| PXX12 | ADP | LRP/LRT | | 2 |
| PXX27 | PDH | PDW | | 1 |
| PXX28 | PMP | PAW/PKW | | 2 |
| PXX45 | PD | DW | 1 | |
| CP06 | VRP | LRT/INR | | 2 |
| CP10 | IMP | IRP | | 1 |
| CP25 | PSM | LRP/MYP | | 2 |
| CP33 | PKL | PRM/PRY/PKW | | 3 |
| KP02 | DI | PAN/PEW | | 2 |
| KP09 | RP | LRA | | 1 |
| KP22 | DP | PEW/EW | 1 | 1 |
| summary | | | 2 | 19 |

B

| Tested | Expressed well | Monomeric | Binding to the designed target by yeast surface display | Affinity < uM by yeast surface display & Octet, BLI |
|---|---|---|---|---|
| 54 | 42 | 25 | 19 | 16 |

**a**, First-round experimental characterization summary. It is clearly shown that among the binders, most of them bound peptides with sequences similar to those targeted but not the same; and peptides with three-residue repeat units were targeted more successfully (19 in total) than those with two-residue repeat units (2 in total). **b**, Second-round experimental characterization summary. In total, 54 second-round designed protein–peptide pairs were tested. Forty-two of the designed proteins were solubly expressed in *E. coli*, 25 were monomerically dispersed by SEC and 16 bound their targets with considerably higher affinity and specificity than in the first round.

**Extended Data Table 2 | Interface side-chain heavy-atom RMSD calculations and SAXS Vr calculations**

| name | RPB_PLP1 _R6-PLPx6 | RPB_LRP2 _R4-LRPx4 | RPB_PLP3 _R6-PLPx6 | RPB_PEW3 _R4-PAWx4 |
|---|---|---|---|---|
| full_iRMSD | 1.61 | 1.43* | 6.49 | 12.41 |
| inter_iRMSD | 1.34 | 1.22* | 4.08 | 9.27 |

A

| Peptide binder | PLPx6 | LRPx6 | PEWx6 | IYPx6 | PRMx6 | PKWx6 |
|---|---|---|---|---|---|---|
| Vr | 1.25 | 0.58 | 1.30 | 1.76 | 1.13 | 0.58 |

B

**a**, Interface side-chain heavy-atom RMSD calculation for four co-crystal complexes and design models. The interface heavy-atom RMSD calculations using Pymol align with cycles=0 (iRMSD for short) was applied to all four crystal-design complexes. For the first-round designs, for example, the values are averaged over five top designs for RPB_PEW3_R4–PAWx4, because the design models were not fully converged (as stated in the main text). For RPB_LRP3_R4–LRPx4, because the final two models were sampling two distinct arginine rotamers as stated in the main text, we calculated the iRMSD for these two, respectively. The closest one was shown above with asterisks, and the further one as (full_iRMSD=5.29, inter_iRMSD=5.16). For all four pairs, we inspected both the full-repeat RMSD with internal-repeat RMSD (N-terminal and C-terminal caps excluded) here, owing to the potential lever-arm effect. **b**, Structural validation of six-repeat peptide binders by SAXS volatility ratio (Vr) calculation.

Daniel Adriano Silva
Emmanuel Derivery

# Reporting Summary

## Statistics

For all statistical analyses, confirm that the following items are present in the figure legend, table legend, main text, or Methods section.

| n/a | Confirmed | |
|---|---|---|
| ☐ | ☒ | The exact sample size (*n*) for each experimental group/condition, given as a discrete number and unit of measurement |
| ☐ | ☒ | A statement on whether measurements were taken from distinct samples or whether the same sample was measured repeatedly |
| ☒ | ☐ | The statistical test(s) used AND whether they are one- or two-sided<br>*Only common tests should be described solely by name; describe more complex techniques in the Methods section.* |
| ☐ | ☒ | A description of all covariates tested |
| ☐ | ☒ | A description of any assumptions or corrections, such as tests of normality and adjustment for multiple comparisons |
| ☐ | ☒ | A full description of the statistical parameters including central tendency (e.g. means) or other basic estimates (e.g. regression coefficient) AND variation (e.g. standard deviation) or associated estimates of uncertainty (e.g. confidence intervals) |
| ☒ | ☐ | For null hypothesis testing, the test statistic (e.g. *F*, *t*, *r*) with confidence intervals, effect sizes, degrees of freedom and *P* value noted<br>*Give P values as exact values whenever suitable.* |
| ☒ | ☐ | For Bayesian analysis, information on the choice of priors and Markov chain Monte Carlo settings |
| ☒ | ☐ | For hierarchical and complex designs, identification of the appropriate level for tests and full reporting of outcomes |
| ☒ | ☐ | Estimates of effect sizes (e.g. Cohen's *d*, Pearson's *r*), indicating how they were calculated |

*Our web collection on statistics for biologists contains articles on many of the points above.*

## Software and code

Policy information about availability of computer code

| Data collection | Microscopy data was collected using Metamorph software v7.10.1.161.<br>Rosetta Macromolecular Modeling Suit; |
|---|---|
| Data analysis | Image analysis was performed using Fiji (ImageJ version: 1.53f). Mass Spectrometry data was analyzed using Scaffold.<br>Python 3.8; ForteBio Data Analysis Software Version 9.0.0.14; FlowJo v10.6.2; Phenix-1.19.2; DNAWorks2.0. |

For manuscripts utilizing custom algorithms or software that are central to the research but not yet described in published literature, software must be made available to editors and reviewers. We strongly encourage code deposition in a community repository (e.g. GitHub). See the Nature Portfolio guidelines for submitting code & software for further information.

## Data

Policy information about availability of data

All manuscripts must include a data availability statement. This statement should provide the following information, where applicable:
- Accession codes, unique identifiers, or web links for publicly available datasets
- A description of any restrictions on data availability
- For clinical datasets or third party data, please ensure that the statement adheres to our policy

The mass spectrometry proteomics has been deposited to the ProteomeXchange Consortium via the PRIDE partner repository with the dataset identifier PXD038492 and 10.6019/PXD038492. The atomic coordinates and experimental data of RPB_PEW3_R4-PAWx4, RPB_PLP3_R6-PLPx6, RPB_LRP2_R4-LRPx4,

RPB_PLP1_R6-PLPx6, RPB_PLP1_R6-PLPx6 (alternative conformation 1), RPB_PLP1_R6-PLPx6 (alternative conformation 2) and RPB_LRP2_R4 (pseudopolymeric) have been deposited in the RCSB PDB with the accession numbers 7UDJ, 7UE2, 7UDK, 7UDL, 7UDM, 7UDN, and 7UDO respectively.
All other data supporting the findings of this study are available from the corresponding authors on reasonable request.

## Human research participants

Policy information about studies involving human research participants and Sex and Gender in Research.

| | |
|---|---|
| Reporting on sex and gender | N/A |
| Population characteristics | N/A |
| Recruitment | N/A |
| Ethics oversight | N/A |

Note that full information on the approval of the study protocol must also be provided in the manuscript.

## Field-specific reporting

Please select the one below that is the best fit for your research. If you are not sure, read the appropriate sections before making your selection.

☒ Life sciences    ☐ Behavioural & social sciences    ☐ Ecological, evolutionary & environmental sciences

For a reference copy of the document with all sections, see nature.com/documents/nr-reporting-summary-flat.pdf

## Life sciences study design

All studies must disclose on these points even when the disclosure is negative.

| | |
|---|---|
| Sample size | 30-60 designs were ordered for each batch of experimental characterization. No statistical method was used to determine the total number. |
| Data exclusions | No data was excluded. |
| Replication | Experimental finders were statistically significant and no attempt at reproduction was performed. |
| Randomization | Randomization was not relevant/not performed. |
| Blinding | Researchers were not blinded/not necessarily blind to perform all the binding assays. |

## Reporting for specific materials, systems and methods

We require information from authors about some types of materials, experimental systems and methods used in many studies. Here, indicate whether each material, system or method listed is relevant to your study. If you are not sure if a list item applies to your research, read the appropriate section before selecting a response.

### Materials & experimental systems

| n/a | Involved in the study |
|---|---|
| ☐ | ☒ Antibodies |
| ☐ | ☒ Eukaryotic cell lines |
| ☒ | ☐ Palaeontology and archaeology |
| ☒ | ☐ Animals and other organisms |
| ☒ | ☐ Clinical data |
| ☒ | ☐ Dual use research of concern |

### Methods

| n/a | Involved in the study |
|---|---|
| ☒ | ☐ ChIP-seq |
| ☐ | ☒ Flow cytometry |
| ☒ | ☐ MRI-based neuroimaging |

### Antibodies

| | |
|---|---|
| Antibodies used | TOM20 antibody (Santa Cruz sc-17764, used it at 1:200 dilution), combined with anti-mouse Alexa Fluor 488 (Invitrogen, A21202 1:500e) for immunofluorescence. Rabbit anti-ZFC3H1 (Sigma, HPA007151, used at 1:250) combined with goat anti-Rabbit Alexa 555 (Invitrogen, A32732, 1:2000) for western blot. Mouse anti-alpha tubulin 488 (Clone DMA1, Sigma T6199, directly labelled with Abberior® STAR 488, NHS ester leading to a 4.5 dye/antibody degree of labelling, and used at 0.1 µg/mL final concentration) for |

| | western blot. |
|---|---|
| Validation | Monoclonal DM1A antibody anti alpha tubulin has been characterized by the manufacturer (https://www.sigmaaldrich.com/GB/en/product/sigma/t6199) . TOM20 antibody has been characterized by the manufacturer (https://datasheets.scbt.com/sc-17764.pdf). |

# Eukaryotic cell lines

Policy information about cell lines and Sex and Gender in Research

| Cell line source(s) | U2OS FlipIn Trex Cells were a kind gift from Stephen C. Blacklow. HeLa FlpIn Trex cells were a kind gift of Simon Bullock |
|---|---|
| Authentication | Cell line used was not authenticated |
| Mycoplasma contamination | Dapi staining did not reveal the presence of Mycoplasms |
| Commonly misidentified lines (See ICLAC register) | No commonly misidentified lines were used |

# Flow Cytometry

## Plots

Confirm that:

☒ The axis labels state the marker and fluorochrome used (e.g. CD4-FITC).

☒ The axis scales are clearly visible. Include numbers along axes only for bottom left plot of group (a 'group' is an analysis of identical markers).

☐ All plots are contour plots with outliers or pseudocolor plots.

☐ A numerical value for number of cells or percentage (with statistics) is provided.

## Methodology

| Sample preparation | EBY100 yeast cells were used for yeast surface display and subjected to the flow cytometry assay. The expression of protein is labeled by FITC conjugated anti-cMyc antibody, while the target is biotinlyated and labeled by SAPE. The samples are prepared and washed in PBS with 1% BSA |
|---|---|
| Instrument | SONY SH800 sorter and Attune NxT Flow Cytometer |
| Software | Softwares of the instruments and python |
| Cell population abundance | The collected yeast cell populations will be grown up in CTUG culture again, and if there is contamination, the cells won't grown well. In most of case, we will also inoculate some liquid yeast cells onto CTUG plates, and check single colony samples to make sure the purify of the cell and check the single colony sequences. |
| Gating strategy | Usually, there will be a control sort, in which there are no target presented. The gating is determined by the control sort. |

☐ Tick this box to confirm that a figure exemplifying the gating strategy is provided in the Supplementary Information.

