## [Peer Review File · Nature]

Manuscript Title: *De novo* design of modular peptide binding proteins by superhelical matching

Reviewer Comments & Author Rebuttals

Reviewer Reports on the Initial Version:

Referees' comments:

Referee #1:

As first structure prediction and then design move into the age of deep learning, it's wonderful to see challenging problems solved using good-old-fashioned chemical principles and some well-designed data structures. This work from the Baker group describes a clever solution to the binding of extended - polyproline conformation repeat sequences, using a protein with matching helical dimensions. The matching of rise and twist parameters, the use of sidechain-backbone hydrogen bonding networks and the application of geometric hashing to match binding sites to protein scaffolds make for an impressive series of designs. The design approach is clearly described. The characterization is thorough - with amazing affinities in the nanomolar to picomolar (!!) range - and a high degree of specificity. Matching of designs and structures show the core approach is sound. I did not find any issues with this work or its presentation.

Referee #2:

This is another excellent piece of science from the Baker group. Effectively, they use rounds of parametric computational protein design and experiment to generate pairs of synthetic receptor (helical repeat) proteins and cognate synthetic ligands (repeat peptides). The underlying idea borrows very much from the excellent work from the Plueckthun lab over the past two decades in developing DARPINs and similar scaffolds by consensus design and then selection to deliver peptide-binding synthetic proteins. The paper under review acknowledges this fully. For me, this foregoing work does take some of the novelty away from the new paper. Nonetheless, the computational approach that the Baker lab brings to this problem is new and very impressive.

Technically—as with many of the papers from this group—the new work cannot be faulted. The study is very well-conceived and well-executed. The computationally designed synthetic protein-peptides pairs have been fully characterized experimentally through X-ray crystal structures and demonstrated to work as pairs in cells. As far as I can tell, both the computational and experimental work are technically sound; indeed, they have been conducted to a very high standard.

The only general question that I have—and this is a decision for the editors at Nature—is that despite the above I do wonder if the impact of this work will come later when the methods are applied to endogenous proteins? At present, the helical-repeat proteins and the repeat-peptide sequences that they target are both designed/synthetic. It would have been more impressive if the

designed proteins targeted natural proteins in cells. For me, this reduces the impact. Indeed—and despite the truly impressive concept and computational work and the volume of experimental work described—I question whether this paper would be better suited to Nature Chemical Biology or Communications at this stage.

Whatever the Editors' decision, the following changes are needed to the current manuscript:

In the first paragraph, the authors comment on the generally low-affinity peptide-protein interactions observed for TPR proteins. I agree with this. However, I note that high-affinity binders are possible and have been achieved through peptide design (Cross et al. (2021) Cell Chem. Biol. 28, 1347-1355). Given that it is related to the authors' own goals, this paper should be mentioned and cited.

I like the computational docking and hash table approach. However, the authors might comment on why they took this approach. That is, rather than say a bioinformatics approach in which all side-chain—side chain and side-chain—mainchain combinations in the PDB are inspected and selected from as knowledge-based templates to slot into the designs.

From their first-generation designs, the authors seem surprised that 3-residue repeat peptides were targeted more successfully than the 2-residue repeats. This seems obvious to me if the targets have PPII conformations, as this has ≈ 3 residues per helical turn. Perhaps I'm missing something, but could the authors develop their comments in the text?

In Fig. 2, the plots in c and d must have complete x and y labels, e.g. the numerical values on each axis in each plot. The values for c are given in the legend, but not for d, which should be wavelength and MRE.

In Fig. 2c, and despite the associated descriptions in the main text, the predicted profiles look very different from the experimental data. This should be expanded on and explained. Also, the experimental SAXS data can be used to generate objects for direct comparison with the computational models; i.e., the process described in the paper can be done the other way around. Has this been done?

In Fig. 2d some of the 20 °C “recovery” spectra do not overlay with the pre-melt spectra. This should be commented on.

In Fig. 3c,e: I recognize that mitochondria can be quite filamentous depending on cell type and conditions. However, they look a little more stringy than usual. Therefore, it would be good to see some control cells (that don't express the synthetic constructs) stained for mitochondria.

As far as I can see, most of the RMSD (inconsistently given as rmsd or RMSD in the manuscript) values between the models and experimental structures are C α only. For designs like these where the precision of side-chain—side chain and side-chain—mainchain contacts is stated upfront as a major target/challenge, it is important to see the accuracy with which these have been achieved. Therefore, it would be good to see the all-atom RMSD values at least for the peptide-protein

contacting regions, and for the authors to comment on the accuracy of their designs at this level.

Finally, the manuscript needs a thorough proofread. At present, it has many inconsistencies in how ranges of numbers, spaces between numbers and units, and so on, are presented.

Author Rebuttals to Initial Comments:

Referee #1:

As first structure prediction and then design move into the age of deep learning, its wonderful to see challenging problems solved using good-old fashioned chemical principles and some well-designed data structures. This work from the Baker group describes a clever solution to the binding of extended - polyproline conformation repeat sequences, using a protein with matching helical dimensions. The matching of rise and twist parameters, the use of sidechain-backbone hydrogen bonding networks and the application of geometric hashing to match binding sites to protein scaffolds make for an impressive series of designs. The design approach is clearly described. The characterization is thorough - with amazing affinities in the nanomolar to picomolar (!!) range - and a high degree of specificity. Matching of designs and structures show the core approach is sound. I did not find any issues with this work or its presentation.

We thank the reviewer for his/her positive feedback on our work!

Referee #2:

This is another excellent piece of science from the Baker group. Effectively, they use rounds of parametric computational protein design and experiment to generate pairs of synthetic receptor (helical repeat) proteins and cognate synthetic ligands (repeat peptides). The underlying idea borrows very much from the excellent work from the Plueckthun lab over the past two decades in developing DARPINs and similar scaffolds by consensus design and then selection to deliver peptide-binding synthetic proteins. The paper under review acknowledges this fully. For me, this foregoing work does take some of the novelty away from the new paper. Nonetheless, the computational approach that the Baker lab brings to this problem is new and very impressive.

Technically—as with many of the papers from this group—the new work cannot be faulted. The study is very well-conceived and well-executed. The computationally designed synthetic protein-peptides pairs have been fully characterized experimentally through X-ray crystal structures and demonstrated to work as pairs in cells. As far as I can tell, both the computational and experimental work are technically sound; indeed, they have been conducted to a very high standard.

The only general question that I have—and this is a decision for the editors at Nature—is that despite the above I do wonder if the impact of this work will come later when the methods are applied to endogenous proteins? At present, the helical-repeat proteins and the repeat-peptide sequences that they target are both designed/synthetic. It would have been more impressive if the designed proteins targeted natural proteins in cells. For me, this reduces the impact. Indeed—and despite the truly impressive concept and computational work and the volume of experimental work described—I question whether this paper would be better suited to Nature Chemical Biology or Communications at this stage.

As noted above, we have spent the last 4-5 months working to address this concern, with quite exciting success as reported in the new final figure in the manuscript and accompanying new paragraph in the main text.

Whatever the Editors' decision, the following changes are needed to the current manuscript:

In the first paragraph, the authors comment on the generally low-affinity peptide-protein interactions observed for TPR proteins. I agree with this. However, I note that high-affinity binders are possible and have been achieved through peptide design (Cross et al. (2021) Cell Chem. Biol. 28, 1347-1355). Given that it is related to the authors' own goals, this paper should be mentioned and cited.

We have added this reference to the manuscript.

I like the computational docking and hash table approach. However, the authors might comment on why they took this approach. That is, rather than say a bioinformatics approach in which all side-chain—side chain and

side-chain—mainchain combinations in the PDB are inspected and selected from as knowledge-based templates to slot into the designs.

We chose to augment the PDB derived statistics with Monte Carlo sampling to more thoroughly cover the space of possibilities—the native orientations alone are relatively sparse leading to missing of many plausible pairing arrangements. We have clarified this in the revised manuscript.

From their first-generation designs, the authors seem surprised that 3-residue repeat peptides were targeted more successfully than the 2-residue repeats. This seems obvious to me if the targets have PPII conformations, as this has ≈ 3 residues per helical turn. Perhaps I'm missing something, but could the authors develop their comments in the text?

We have reworded this portion of the text to make it clearer.

In Fig. 2, the plots in c and d must have complete x and y labels, e.g. the numerical values on each axis in each plot. The values for c are given in the legend, but not for d, which should be wavelength and MRE.

Thanks for pointing this out—we have fixed this in the revised manuscript.

In Fig. 2c, and despite the associated descriptions in the main text, the predicted profiles look very different from the experimental data. This should be expanded on and explained. Also, the experimental SAXS data can be used to generate objects for direct comparison with the computational models; i.e., the process described in the paper can be done the other way around. Has this been done?

We have carried out a more quantitative analysis of the fit between the experimental and predicted spectra using the volatility ratio statistic V_r . Previous studies of repeat protein designs have found that crystallographically validated designs can have V_r values up to 2.5 ; as shown in the table below (now added to the SI), the values for all designs were well below this threshold. We prefer not to generate objects from SAXS data with relatively few features as the experimental data are not very information rich.

V_r value calculated as follows:

Peptide binder	PLPx6	LRPx6	PEWx6	IYPx6	PRMx6	PKWx6
V_r	1.25	0.58	1.30	1.76	1.13	0.58

In Fig. 2d some of the 20 °C “recovery” spectra do not overlay with the pre-melt spectra. This should be commented on.

This is correct—about half of the designs don't fully recover at 20C. A small fraction of the proteins likely crash out at 95C due to the exposed surface hydrophobics. We have noted this in the revised manuscript.

In Fig. 3c,e: I recognize that mitochondria can be quite filamentous depending on cell type and conditions. However, they look a little more stringy than usual. Therefore, it would be good to see some control cells (that don't express the synthetic constructs) stained for mitochondria.

We thank the reviewer for pointing this out. The reviewer is right that mitochondria shape can greatly vary between cell types, but also the cell state, such as mitosis, migration, spreading.... In our experiments, we always image spreading cells onto fibronectin-coated glass to ensure optimal imaging conditions, rather than cells let to adhere on glass overnight. This may well be the reason why the mitochondria signal appeared more "stringy" than usual in Fig.3, but, as correctly pointed out by the reviewer, it could also have been because our construct was affecting mitochondria shape. We thus performed the experiment suggested by the reviewer and did endogenous TOM20 immunofluorescence in wild-type U2OS cells to spread onto fibronectin-coated glass for the same amount of time. Mitochondria looked very similar in these controls compared to the cells overexpressing the mitochondria-targeted binders, suggesting that cell spreading is the main reason why our mitochondria look the way they look. This data is presented in new Fig.S7, and we added in the legend of Fig.3 that the cells were spreading.

As far as I can see, most of the RMSD (inconsistently given as rmsd or RMSD in the manuscript) values between the models and experimental structures are C α only. For designs like these where the precision of side-chain—side chain and side-chain—mainchain contacts is stated upfront as a major target/challenge, it is important to see the accuracy with which these have been achieved. Therefore, it would be good to see the all-atom RMSD values at least for the peptide-protein contacting regions, and for the authors to comment on the accuracy of their designs at this level.

This is an excellent suggestion. We computed the all atom interface RMSD over both the full complex or the middle four repeats (because of deviations in the capping repeats) for all four crystal-design complexes, and added a supplemental table with these data to the revised manuscript. The sidechain RMSD was less than 2.0 for the second round designs, but considerably higher for the first round designs, consistent with the greater deviations in the backbone.

Interface side-chain RMSD calculated as follows:

name	RPB_PLP1 _R6-PLPx6	RPB_LRP2 _R4-LRPx4	RPB_PLP3 _R6-PLPx6	RPB_PEW3 _R4-PAWx4
full_iRMSD	1.61	1.43*	6.49	12.41
inter_iRMSD	1.34	1.22*	4.08	9.27

Finally, the manuscript needs a thorough proofread. At present, it has many inconsistencies in how ranges of numbers, spaces between numbers and units, and so on, are presented.

We have gone through and tried to fix all the inconsistencies—sorry about this!

Reviewer Reports on the First Revision:

Referees' comments:

Referee #2 (Remarks to the Author):

First of all, I apologize to the authors and editors that it has taken me a while to turn this second review around – it has been an extraordinarily busy time for me.

I am pleased that the authors have gone to some lengths to address all of my comments on the original submission. The revised paper is excellent, and I feel that it can now be published in Nature.

The new work targeting the endogenous ZFC3H paper is a very good and welcome addition, which, in my view, raises both the paper and the bar in protein design higher. One point here: from Fig. 6C, the binding achieved to the natural ligand is at $\approx\mu\text{M}$ or slightly better. The authors should state the number in the text, and comment on this compared with designed pairings from earlier in the paper, and the use of such a relatively modest binder in cell biology.

On this more generally—though I recognize that the approach and target are orthogonal to the approach taken by Wu et al. and Baker—I see that the authors of reference 15 (which is good to see cited now) have since published a follow-on paper in Nature Chemical Biology. This shows that their designer peptide can be introduced as an exogenous reagent to target an endogenous motor protein. As I say, it is analogous to what the authors have done rather than being directly related to it, but it would be worth citing this paper too, as the authors' paper and the Nat. Chem. Biol. paper show the new and exciting directions that peptide/protein design is going in for sub-cellular applications.

Rhys, G. G. et al. De novo designed peptides for cellular delivery and subcellular localization. Nat. Chem. Biol. 18, 999-1004, DOI: 10.1038/s41589-022-01076-6 (2022)

The authors have also addressed most of the other questions that I raised quite well. However, there are still quite a few typos and inconsistencies that need correcting. Here are a few:

Lines 210/211 and throughout the MS – 95°C, 20C, 95C.

Lines 262/264 and after – RMSD and rmsd

Line 351 – Fig. 5c should be 6c, presumably

Throughout the MS amino acid full names do not need to be capitalized – they are not proper nouns. However, the three-letter codes do – see line 677 “his-pull down” should be “His-pull-down”, for example.

Author Rebuttals to First Revision:

Referees' comments:

Referee #2 (Remarks to the Author):

First of all, I apologize to the authors and editors that it has taken me a while to turn this second review around – it has been an extraordinarily busy time for me.

I am pleased that the authors have gone to some lengths to address all of my comments on the original submission. The revised paper is excellent, and I feel that it can now be published in Nature.

We thank the reviewer for the acknowledgement and support of our work!

The new work targeting the endogenous ZFC3H paper is a very good and welcome addition, which, in my view, raises both the paper and the bar in protein design higher. One point here: from Fig. 6C, the binding achieved to the natural ligand is at $\approx\mu\text{M}$ or slightly better. The authors should state the number in the text, and comment on this compared with designed pairings from earlier in the paper, and the use of such a relatively modest binder in cell biology.

Sorry for the missing information - we added it back to the text. Both FP fitting and Octet kinetics fitting gave ~ 200 nM Kd for the αZFC -high binder. We understand there seemed to be some non-specific binding at highest concentration ($>2 \mu\text{M}$). Note that as a proof of concept, we only made 8 designs in total (compared to dozens of designs for the earlier synthetic pairs), and moved the best one forward to cell biology since we reasoned that even a 200 nM affinity binder would be sufficient to pull down the native PAXT complex when bound to a resin because of an avidity effect (and in fact, it worked as intended). It is likely that with further engineering beyond the scope of this proof of concept, a higher affinity can be reached.

When it turns to applications, even binders with modest affinity (i.e. in the 100 nM range) are already a game changer in cell biology (Cheloha et al. JCB 2020). First, classic applications involve some kind of avidity boost by concentrating the binder onto a given surface. This is what happens for affinity purification of native multiprotein complex using resin-bound binders, as we did here (Fig. 6), but also in so called “knock sideways” experiments, where a binder is concentrated onto a target organelle (e.g. mitochondria) to relocalize an endogenous target to it (Fig.3). Furthermore, a binder toward the sequence EPEA with similar affinity to αZFC -high ($190 \mu\text{M}$) has been successfully used to target enzymes to EPEA-tagged proteins to glycosylate them in cells (Ramirez et al. ACS Chem. Biol. 2020). Last, high affinity binders are actually not always desired for applications where binders are fused to fluorescent proteins to image the dynamics of the target in cells because there is always the risk that the binder affects the activity of the target, making it a bad probe. A classic example of this is the lifeact probe for actin filaments, which has a modest affinity ($1.2 \mu\text{M}$), but that is the good

tradeoff not to perturb actin dynamics (Kumari et al, EMBO J, 2020). Our technology paves the way towards applying all these techniques to *endogenously* expressed proteins rather than exogenously overexpressed tagged proteins, which is often problematic, especially when working with multiprotein complexes.

On this more generally—though I recognize that the approach and target are orthogonal to the approach taken by Wu et al. and Baker—I see that the authors of reference 15 (which is good to see cited now) have since published a follow-on paper in Nature Chemical Biology. This shows that their designer peptide can be introduced as an exogenous reagent to target an endogenous motor protein. As I say, it is analogous to what the authors have done rather than being directly related to it, but it would be worth citing this paper too, as the authors' paper and the Nat. Chem. Biol. paper show the new and exciting directions that peptide/protein design is going in for sub-cellular applications.

Rhys, G. G. et al. De novo designed peptides for cellular delivery and subcellular localization. Nat. Chem. Biol. 18, 999-1004, DOI: 10.1038/s41589-022-01076-6 (2022)

Exciting new paper- We added it to our reference!

The authors have also addressed most of the other questions that I raised quite well. However, there are still quite a few typos and inconsistencies that need correcting. Here are a few:

Lines 210/211 and throughout the MS – 95°C, 20C, 95C.

Lines 262/264 and after – RMSD and rmsd

Line 351 – Fig. 5c should be 6c, presumably

Throughout the MS amino acid full names do not need to be capitalized – they are not proper nouns. However, the three-letter codes do – see line 677 “his-pull down” should be “His-pull-down”, for example.

We apologize again for the inconvenience- fixed it accordingly.